

# Identification of ice-over-water multilayer clouds using multispectral satellite data in an artificial neural network

Sunny Sun-Mack[1], Patrick Minnis[1], Yan Chen[1], Gang Hong[1], and William L. Smith, Jr.[2]

[1]Analytical Mechanics Associates, Inc., Hampton, VA, USA 23666

[2]NASA Langley Research Center, Hampton, VA, USA 23681

*Correspondence to*: Sunny Sun-Mack (szedung.sun-mack-1@nasa.gov)

**Abstract.** An artificial neural network (ANN) algorithm, employing several Aqua MODerate-resolution Imaging Spectroradiometer (MODIS) channels, the retrieved cloud phase and total cloud visible optical depth, and temperature and humidity vertical profiles is trained to detect multilayer (ML) ice-over-water cloud systems identified by matched 2008 CloudSat and CALIPSO (CC) data. The trained MLANN was applied to 2009 MODIS data resulting in combined ML and single layer detection accuracies of 87% (89%) and 86% (89%) for snow-free (snow-covered) regions during the day and night, respectively. Overall, it detects 55% and ~30% of the CC ML clouds over snow-free and snow-covered surfaces, respectively, and has a relatively low false alarm rate. The net gain in accuracy, which is the difference between the true and false ML fractions, is 7.5% and ~2.0% over snow-free and snow/ice-covered surfaces. Overall, the MLANN is more accurate than most currently available methods. When corrected for the viewing-zenith-angle dependence of each parameter, the ML fraction detected is relatively invariant across the swath. Compared to the CC ML variability, the MLANN is robust seasonally and interannually, and produces similar distribution patterns over the globe, except in the polar regions. Additional research is needed to conclusively evaluate the VZA dependence and further improve the MLANN accuracy. This approach should greatly improve the monitoring of cloud vertical structure using operational passive sensors.

## 1 Introduction

Passive remote sensing with polar-orbiting and geostationary passive imagers is currently the only approach suitable for nearly continuous monitoring of clouds day and night around the globe. While cloud remote sensing is well established and the methodologies are abundant (e.g., Stubenrauch et al., 2013), detecting and characterizing multilayered clouds remains a continuing challenge. Typically, algorithms employed to retrieve properties such as cloud optical depth, effective hydrometeor size, top height, and phase, treat the radiances for a given cloudy imager pixel as emanating from a single plane-parallel cloud sheet. Rarely, if ever, will an actual cloud entirely satisfy the plane-parallel assumptions. Instead, the sizes and densities of the hydrometeors vary vertically and horizontally within the atmospheric column corresponding to a cloudy imager pixel. The top and side surfaces of clouds, even stratus, typically have bumps and troughs that deviate to various degrees from the uniformity



implicit in the plane-parallel model (e.g., Loeb and Coakley, 1998). Simply having vertical extent can cancel the plane-parallel
assumption when viewing parts of a cloud side (e.g., Liou and Ou, 1979). Multilayered clouds also violate the model.
Accounting for variations in single-layer-cloud morphology with a non-plane-parallel type of model is too complex for
use in operational retrieval algorithms and likely requires information that is currently unavailable in most imager radiance
datasets. Moreover, radiative transfer calculations used in weather and climate models are based on the same plane-parallel
premise, although methods are being developed to account for some 3D structure (e.g., Schäfer et al., 2016). For single-layer
(SL) clouds, the nonuniform geometry is the predominant deviation from the plane-parallel ideal. It mostly affects retrievals
of cloud optical depth (COD) and particle effective radius (CER), but, less so, cloud-top phase and height (CTH). The presence
of two water phases and separation of the upper and lower layers in ice-over-water multilayer (ML) systems can also produce
large errors in COD and CER, and significantly diminish the accuracies of thermodynamic phase and cloud-top height (CTH)
retrievals (e.g., Minnis et al., 2007; Yost et al., 2023). Reducing uncertainties in the retrievals due to nonconformance with the
SL plane-parallel ideal, particularly for ML clouds, is critical to increasing the value of imager cloud retrievals for a variety of
applications.
Reliable determination of cloud characteristics is critical to the Clouds and the Earth's Radiant Energy System (CERES,
Loeb et al., 2016) for converting broadband radiance measurements to reliable shortwave and longwave fluxes at the top of
the atmosphere, within the atmosphere, and at the surface. Cloud properties extracted from satellite imagery are also exploited
in a wide variety of applications. These include, among others, verifying climate model cloud parameters (e.g., Zhang et al.,
2010; Stanfield et al., 2014), enhancing aviation safety (Mecikalski et al., 2007; Smith et al., 2012), and improving short-term
weather forecasts (e.g., Kurzock et al., 2019; Benjamin et al., 2021) and estimating surface radiative fluxes (e.g., Rutan et al.,
2015; Ryu et al., 2018). All of these applications and others (e.g., Chen and Zhang, 2000; Morcrette and Christian, 2000) will
benefit from more accurate cloud properties, especially for ML systems.
Active instruments such as the Cloud-Aerosol Lidar with Orthogonal Polarization (CALIOP) lidar on the Cloud-Aerosol
Lidar and Infrared Pathfinder Satellite Observations (CALIPSO) satellite (Winker et al., 2009) and the Cloud Penetrating
Radar (CPR) on CloudSat (Stephens et al., 2008) together have produced quite accurate depictions of cloud vertical structure.
These satellites are part of the A-Train, the Afternoon Constellation of sun-synchronous orbiters that, for years, flew nearly
the same tracks (13:30 Equatorial crossing time) separated by only a few minutes. Other satellites with imagers, particularly
Aqua with the MODerate-resolution Imaging Spectroradiometer (MODIS), are also members of the A-Train. The CALIOP
and the CPR are both near-nadir viewing instruments that generate profiles of atmospheric particles only in a narrow curtain
along the satellite track. Those profiles, which often include overlapping clouds, are valuable for many uses, especially when
combined with other instruments on the A-Train. Because they sample only a tiny fraction of the globe at two local times each
day, however, the current active instruments have limited utility for many of the applications served by operational satellite
imager products.



Efforts to accurately identify and unscramble ML clouds from passive imagery have yielded a variety of methods that
have either been demonstrated as efficacious or are being applied routinely. They are based on interpreting radiances from
either multiple instruments or from a single instrument with multiple channels. To identify ML clouds, Lin et al. (1998)
matched microwave radiometer (MWR) retrievals of cloud liquid water path (LWP) from polar-orbiting satellites with
retrievals of COD from geostationary satellites. Minnis et al. (2007) used MWR retrievals of LWP matched with imager
retrievals of COD and CER to detect and retrieve ML cloud properties over water surfaces. By combining Aqua MODIS CTP
and COD with the optical centroid cloud pressure retrieved from the Ozone Monitoring Instrument on the Aura satellite, *Joiner*
*et al.* [2010] discriminated between vertically extended and ML clouds.
The single-instrument approach, which is more viable for monitoring ML clouds from a greater number of satellites, often
relies on discrepancies between the visible (VIS, ~0.65 μm) channel COD and that determined from other channels. For
example, COD derived from infrared radiances is limited to values of less than ~5 because the usable signal diminishes for
thicker clouds. Thus, the two COD retrievals can be used to detect thin cirrus over a thicker lower cloud. Pavolonis and
Heidinger (2004) identified ML clouds by comparing the COD retrieved from the brightness temperature differences from the
11 and 12-μm channels, $BTD_{1112}$, with the VIS COD. They suggested that the MODIS 1.38-μm and 1.63-μm reflectances
could be combined with $BTD_{1112}$ to retrieve the cirrus optical depths for comparison with the VIS COD. The MODIS $CO_2$
channels were used by *Chang and Li* [2005] to retrieve IR COD for ML cloud detection. Their method was simplified by
*Chang et al.* [2010] to employ brightness temperatures from $CO_2$ and 11-μm channels to identify high clouds and ultimately
to detect ML clouds in the CERES MODIS Edition 4 products (CM4; Minnis et al., 2021). Similarly, *Wind et al.* [2010]
contrasted the cloud top pressure (CTP) derived with a $CO_2$ method to that based on absorption in the MODIS 0.94-μm channel
along with other tests to identify ML clouds in MODIS pixels. To further improve ML detection, this technique was enhanced
with additional tests, including the Pavolonis and Heidinger (2002) method (Platnick et al., 2017; Marchant et al., 2020).
Desmons et al. (2017) exploited multi-angle polarized spectral reflectances and two different retrievals of CTP from the
Polarization and Directionality of the Earth's Reflectance (POLDER) instrument on another A-Train satellite. Instead of using
COD retrievals, Wang et al. (2019) utilized a series of tests applied to three spectral reflectances and two infrared brightness
temperatures measured by the Visible Infrared Imaging Radiometer Suite (VIIRS) to classify clouds as single-layer ice or
water, multilayer, probable multilayer, or uncertain phase and layering. Their technique yielded results similar to those from
Platnick et al. (2017). For detecting ML clouds in VIIRS data, CERES replaced the $CO_2$ channel with the 12-μm channel in
the Chang et al. (2010) approach. It found fewer ML clouds than the method using the $CO_2$ channel (Minnis et al. 2023).
CERES is focused on accurately characterizing the radiation budget at the top of the atmosphere, at the surface, and within
the atmosphere, so it is important to reliably identify all clouds and retrieve their properties, ideally at all levels within the
atmosphere. To improve the CERES ML detection, Sun-Mack et al. (2017) began the development of an artificial neural
network (ANN) to distinguish between SL and ML clouds using MODIS radiance data matched to CALIPSO and CloudSat
vertical profiles of clouds.  Minnis et al. (2019) further enhanced this multilayer ANN (MLANN) by including more input



parameters and additional output variables such as upper layer CTH, COD, and cloud-base height (CBH). They also used only
high-confidence CloudSat and CALIPSO data for training. They found that, for nonpolar clouds, the MLANN correctly
identified ML and SL clouds together 80.4% and 77.1% of the time during day and night, respectively, using CALIPSO data
averaged over an 80-km distance. While those results are quite encouraging, the approach needs further refinement and global
coverage. Tan et al. (2022) applied several machine-learning methods to a geostationary satellite dataset and found even greater
accuracy with a random-forest technique. It is clear that artificial intelligence techniques are the most promising approach to
detecting ML clouds from passive imagery.

This paper reports on continued development of the MLANN to detect ML clouds. Revisions to the previous training are
made using a newer version of CALIPSO products with constrained horizontal resolution. In addition to the separate day/night
training used in previous versions, the MLANN herein is trained separately for CERES ice and water cloud pixels separately
for snow-free and snow/ice-covered surfaces. Finally, because the MLANN is trained with near-nadir data, its utility for full
swath MODIS data is examined.
**2 Artificial neural network**
A neural network is a form of artificial intelligence used to identify a relationship between input and output variables by
learning from a set of observed or simulated data (e.g., Karayiannis and Venetsanopoulus, 1993). Neural networks are
increasingly used in cloud remote sensing using satellite imagery. Kox et al. (2014) and Strandgren et al. (2017) employed an
ANN to determine cirrus COD and CTH, while Cerdena et al. (2007) used it to estimate liquid water cloud COD and CER and
Taravat et al. (2015) detected the presence of clouds with it. Using an ANN, Minnis et al. (2016) retrieved thick ice cloud
COD at night and Håkansson et al. (2018) accurately determined CTP and CTH. Stengel et al. (2020), Wang et al. (2020), and
White et al. (2021) use ANNs for cloud detection and phase discrimination. These and other examples demonstrate that ANNs
have significant potential for improving the characterization of global cloudiness. This potential appears to derive primarily
from the ability of the method to better interpret several subtle, but often ambiguous radiative signals that are difficult to
reconcile in physically based retrievals. This is particularly true for thick clouds at night (e.g., Minnis et al. 2016) or for ML
clouds. The frequency of ambiguous signals in the radiance fields spurred development of the MLANN by Sun-Mack et al.
(2017).

The MLANN is used to decide if a pixel contains a SL or ML cloud by relating a vector of input data to output (see
schematic in Fig. 1). Each input neuron, the variable, $x_i$, is linked to each member of a hidden layer through the function, $u_j =$
$f1(g_j)$, where $g_j$ is the sum of the weights $w_{ij}$ applied to the input variable $x_i$ plus the constant $b_j$, as indicated Fig. 1. The output
layer is uniquely linked to the input through the hidden layer by the weights $w_j$ and the constant $c$. The number of neurons $j$,
$N_j$, is selected to optimize the output accuracy. Likewise, the number of input variables or $i$ neurons is also adjusted to minimize
the output uncertainty. In this study, 50 -70 neurons are used for the hidden layer. The output is a probability between 0.0 and
1.0.



The Levenberg-Marquadt optimization method is employed to calculate $w_{ij}$ in the MLANN training. The entire process to
solve for the weights consists of three components: training, testing, and validation. A majority of the data is used to compute
the weights, while smaller portions are used for testing and validating the MLANN. Because the testing and training are
interwoven, the testing results force the fitting or training process to stop or to further adjust the weight values. When the
process stops (i.e., no further improvement is gained from adjusting the weights), the training ends, yielding the final weights.
The validation data are then used to determine if their output yields error statistics that are within the maximum stipulated
error. Validation vectors are used to stop training early if the MLANN performance fails to improve. The formulation of this
approach is given in more detail by Minnis et al. (2016, 2019) and Sun-Mack et al. (2017) along with the references found
therein.

## 3 Data and methodology

The MLANN is trained with Aqua MODIS imager data and cloud products, and numerical weather model analyses as input
and active sensor data as the output. Training is performed using 2008 data sampled to accommodate computer memory
capacity and the percentage of ML clouds. The results are assessed using 2008, 2009, and 2013 data. According to Kato et al.
(2010), approximately 51% of cloud systems identified by the CPR and CALIOP consist of two or more layers separated by
at least 200 m. Of these, 57, 28, 10, 4, and 1% have 2, 3, 4, 5, and 6+ layers, respectively. Those statistics include liquid-over-
liquid, ice-over-ice, water-over-ice as well as ice-over-water cloud overlap. Unscrambling this variety of layering is a daunting
task.
To simplify ML detection and later retrievals, only those clouds having the greatest differences in properties are assumed
to be multilayered. Thus, only systems having ice-over-liquid clouds are considered because they differ in phase, scattering
properties, and altitude, and are more common than liquid-over-ice clouds. Thus, multilayered clouds are defined here as any
combination of ice-cloud layers above one or more water cloud layers with the constraint that the top of the uppermost water
layer must be at least 1 km below the bottom of the lowest ice cloud layer. All ice cloud layers together are considered to
constitute a single cloud layer. Similarly, all liquid layers are considered together as a single layer. Selection of 1 km as the
minimum separation distance is based on the need to ensure complete separation between the ice and water layers and to
maximize the number of detected ML clouds. Sun-Mack et al. (2017) and Minnis et al. (2019) found that a larger separation
distance can result in greater accuracy but at the expense of missing a significant number of actual ML clouds. Tan et al. (2022)
have demonstrated that detection accuracy is significantly compromised when the separation distance is less than 1 km.

### 3.1 Data

MODIS on Aqua, the CALIOP, and the CPR, took measurements continuously within ±3 min of each other over a given
location until 2011, when CloudSat suffered battery problems and thereafter only collected data during the day. CloudSat



exited the A-Train during February 2018. The CALIOP and the CPR were aligned to view nearly the same area along their
respective orbits. Because their flight tracks are typically close to the Aqua nadir path, MODIS scans the same scene at viewing
zenith angles, VZA < 18°. Vertical profiles of clouds from the CALIPSO Version 4 (Vaughan et al., 2016) and the CloudSat
2B-CLDCLASS P_R04 (Sassen and Wang, 2008) datasets were matched with 1-km Aqua MODIS Collection 6.1 radiances
and CERES-retrieved cloud properties in an updated CloudSat, CALIPSO, CERES, and MODIS (C3M) product of Kato et al.
(2010). The CERES MODIS cloud properties of interest, cloud-top phase, cloud particle effective radius, $R_{CM}$, and COD, $\tau_{CM}$,
were retrieved from MODIS radiances using an interim CERES Edition 5 algorithm, CM4+. It uses the CM4 algorithms with
two changes. In CM4+, $\tau_{CM}$ is retrieved over snow and ice using a combination of 1.61-μm and 1.24-μm reflectances, as
suggested by Minnis et al. (2021). The former channel is used for thinner clouds (COD < 8 for ice clouds, COD < 32 for water
clouds) and the latter for thicker clouds. The Aqua MODIS 1.6-μm channel consists of 10 sensors. Of those ten, only six
operate properly. To obtain full 1.6-μm imagery, the 1.6-μm reflectance for each bad pixel was replaced with that from the
nearest good detector. That method was applied to the 101-pixel-wide MODIS swath of the C3M. The other change for CM4+
is the use of the two-habit ice crystal model of  Loeb et al. (2018) for retrieving ice cloud properties.
A separate dataset is the full-swath Aqua CM4 cloud product and the Aqua MODIS Collection 6.1 radiances, which are
sampled every other scan line and every fourth pixel. Note that the 1.6-μm channel is only available from 3 of the five sensors
on the CERES sampled Aqua MODIS Collection 6.1 data. The corrections applied to faulty 1.6-μm sensors for C3M have not
yet been implemented for CERES Aqua MODIS data prior to 2019. Therefore, the sampling is reduced and all data from the
faulty sensors is not included in the full-swath 2009 and 2013 data.
To produce a complete vertical profile of cloud-filled layers, the C3M converts the CloudSat CLDCLASS high-confidence
cloud profiles from 240–m to 60-m vertical resolution, then merges them with the CALIPSO cloud profile and vertical feature
mask. The nominal horizontal footprint of a CALIOP shot at the Earth's surface is 330 m in diameter. To detect fainter clouds,
the CALIPSO processing system computes horizontal averages (HA) of the lidar signals from multiple shots corresponding to
increasing distances along the track: 1, 5, 20, and 80 km.
Trepte et al. (2019) established that the daytime CM4 cloud detection rate over ocean dropped from nearly 80% for the
optically thinnest clouds from CALIPSO, those having COD < 0.1, to 50% or less when the clouds detected by the CALIOP
with HA > 20 and 80-km were added to those clouds found using HA ≤ 5 km. This indicates that few cirrus clouds discernible
only at the longer CALIPSO averaging distances can be detected and are less likely to be identified in ML conditions. In the
previous MLANN formulation (Minnis et al. 2019), CALIPSO data with HA ≤ 80 km were used, no $\tau_{CM}$ minimum value was
assigned, and vertical profiles of humidity were not included in the input. Thus, extremely thin cirrus clouds were included in
that earlier training. Only CALIPSO data with HA ≤ 5 km are used here to define the cloud profiles. Additionally, in the
previous analysis, all three CALIPSO 0.33-km pixels matched to a given MODIS pixel were required to be cloudy after the
horizontal averaging was performed. Here, only two out of three 0.33-km pixels are required to be cloudy and any cloud having
$\tau_{CM}$ < 0.5 is assumed to be single-layered. The latter constraint assumes that the ML signal from such optically thin clouds is



negligible, and any retrieval attempt will yield upper and lower cloud-layer properties that are, at most, highly uncertain. To
cover all seasons for snow-free surfaces and facilitate computer processing for training, the C3M data were sampled every
fourth pixel of 2008 for snow/ice-free (SF) areas, while all pixels were used for the snow/ice-covered (SC) scenes. This full-
year training set is more comprehensive than the 1-month dataset of Minnis et al. (2019).
The C3M data were merged with the relevant surface skin temperature and vertical profiles of relative humidity taken
from the CERES Meteorology, Ozone, and Aerosol (MOA) product (Gupta et al., 1997). The latest MOA product is the result
of regridding and interpolating spatially and temporally the version 5.4 reanalysis produced by the Global Modeling
Assimilation Office Global Earth Observing System (GEOS-5.4), an update of the versions described by Rienecker et al.
(2008). These are the same data employed in the CM4 retrievals. To perform additional analyses, the CC COD, $\tau_{CC}$, was
computed for each ice-layer pixel identified as ML in the CALIPSO-CloudSat profiles. The value of $\tau_{CC}$ is equal to the
CALIPSO ice COD, when the CALIOP signal shows a return from the lower layer cloud, otherwise it is equal to the combined
CALIPSO-CloudSat COD.

**3.2 Input and Output Layers**

The MODIS input variables are listed in Table 1. They include latitude, longitude, surface elevation, brightness temperatures
$T_\lambda$ and brightness temperature differences, $BTD_{\lambda 1 \lambda 2} = T_{\lambda 1} - T_{\lambda 2}$, where $\lambda$ is the wavelength in μm abbreviated to the first two
digits. Here, brightness temperatures at 3.7, 6.7, 8.5, 11, 12, and 13.3 μm are used together with $BTD_{3711}$, $BTD_{6711}$, $BTD_{8511}$,
$BTD_{1112}$, and $BTD_{1113}$. The parameters involving the 13.3-μm channel were not used over snow-covered surfaces because of
striping in the 13.3-μm images over Greenland and Antarctica. The GEOS-5.4 input data consist of the surface skin temperature
and relative humidity at the surface and at 850, 700, 500, 400, 300, 200, and 100 hPa. During the day, solar zenith angle (SZA)
and the reflectances, $\rho_\lambda$, from the MODIS 1.38, 1.61, and 2.13 μm channels and the reflectance differences between 1.6 and
2.1 μm. CM4+ retrievals of $\tau_{CM}$ and $R_{CM}$ are also employed for the MLANN formulation. The reflectances and $R_{CM}$ were not
used by Minnis et al. (2019).
A cloudy pixel is considered to be ML, if the MLANN output probability is $\geq$ 0.5. Otherwise, it is identified as SL. Three
components are used in the training process: training, testing, and validation. The sampled 2008 data are split with 60%, 20%,
and 20% each for the respective components. To provide an independent validation of the trained MLANN, a fully sampled
Aqua MODIS dataset from 2009 is analyzed and the results compared with the matching CC data.
A total of eight training sets for each parameter of interest are developed based on the surface condition (snow-free or
snow-covered), time of day and the CM4+ cloud-top phase selection. Night is defined as times when SZA > 82°, while daytime
corresponds to SZA $\leq$ 82°. Each training set yields the weights, $w_{ij}$ and $w_j$, along with the constants $b$ and $c$. Training is
performed using the minimum ice-liquid cloud separation threshold, $\Delta Z_{min} = 1$, where the separation $\Delta Z$ is the difference in
altitude between the bottom of the lowest ice layer and the top of the highest water layer in the CC profile. An ice-cloud layer
is assumed to be present in the profile, if





(1) CALIPSO VFM cloud phase is either ice clouds or mixed phase clouds, or

(2) at least, one layer occurs at a height above the altitude corresponding to 253 K and no temperature
inversion exists in the atmospheric layer between the altitudes corresponding to 273 K and 253 K.

All C3M data having an ice cloud layer by that definition are used as output in the MLANN training. A given profile is assumed to contain a ML cloud system if a cloudy layer exists at least 1 km below the lowest ice cloud layer and the base of that cloud layer is below the 253-K level. Ice-over-water layer systems are considered to be contiguous if they are closer than 1 km. This separation ensures that the two sensors have actually detected a ML cloud system.

Figure 2 breaks down the CC classification according to the CM phase selection, which presumably corresponds to the cloud top. Over SF surfaces, 68% of the MODIS pixels classified as ice during the day (Fig. 2a) are SL ice according to CC, while ~25% correspond to ML clouds. The CM phase is incorrect for ~7% of the cases. At night, more ML clouds are evident and nearly 10% of the ice clouds are classified as liquid. For SF liquid clouds from MODIS (Fig. 2b), the day and night results are comparable with ~15% of the clouds actually being SL ice according to the active instruments. A smaller fraction of the ML clouds correspond to the CM liquid phase than for the ice phase. Yet overall, more ML clouds are found for the CM water phase because the liquid pixels comprise 63% of the $9.7 \times 10^6$ SF observations during the day and 58% of the $11.1 \times 10^6$ SF pixels at night. The breakdown is different for SC scenes. During the day (Fig. 2c), 80% of the ice pixels are SL ice and only 12% are ML. The phase accuracy is reduced at night, but the ML cloud fraction is the same. For CM liquid clouds (Fig. 2d), multi-layer cloud occurrence is slightly greater than for the ice clouds, while phase accuracy is the worst for any of the four categories. Around 30% of the SL ice clouds are mistaken as liquid.

## 4 Results

The results presented here consist of comparisons of the MLANN and corresponding CC parameters for the 2008 training dataset along with data from 2009 to ensure robustness of the estimates. Weights and constants were determined by training for each category and parameter and then applied to the independent datasets. The MLANN was trained using the C3M data for the four categories in Fig. 2 to obtain four sets of weights and constants for each surface type.

### 4.1 Multi-layer cloud detection

Figure 3 plots the CC cloud profiles retrieved over six areas during 25 December, 2009. The CC layering classification uses gray for SL and blue for ML. In addition to the cloud profiles, the MLANN selection of SL (gray) and ML (blue) is indicated by the lines of dots across the top section of each panel. The surface elevation is denoted by the black dashed line in each panel. On the left are daytime observations over the tropical Atlantic (Fig. 3a), eastern Europe (Fig. 3b), and eastern Antarctica (Fig. 3c). Nocturnal profiles are given on the right for passes over the tropical Indian Ocean (Fig. 3d), the south Pacific (Fig.



3e), and northern Russia (Fig. 3f). In the tropical overpasses, the MLANN detects a large fraction of the ML clouds, but also
misses a noticeable number of ML pixels. For example, only a few of the intermittent ML clouds between 2.5°N and 6.1°N
are identified by the MLANN in Fig. 3a and a segment of continuous ML clouds near 12°S in Fig. 3d is missed by the MLANN.
Similar results are seen for the mid-latitude SF areas, where a few ML clouds are missed around 59°N in Fig. 3b and near 35°S
in Fig. 3e.
Some false ML clouds are also found in these panels. In Fig. 3b, for example, false ML clouds are evident at 51°N and
also in two areas between 56 and 58°N. In the latter case, there are ML clouds in the profile but they were not classified as
such by the CC constraints, possibly due to the 253 K liquid cloud temperature threshold. Thus, some of the false ML may
actually be ML clouds. The detection rates for the two SC profiles are much reduced. During the daytime case (Fig. 3c), only
one stretch of ML clouds is detected, while even fewer ML clouds are detected at night around 76°N (Fig. 3f). The unidentified
ML clouds are more common in both cases.
To summarize the results for all of the data, a confusion matrix was constructed for each category. Referring to Table 2,
agreements between the MLANN and CC SL and ML classifications are denoted as SS and MM, respectively, while false SL
and ML pixels are given by SM and MS, respectively. These are used to define the following parameters.
Fraction correct (FC) or Accuracy: $ACC = FC = SS + MM$. (1)
Real Risk: $RR = SM + MS = 1 - ACC$. (2)
False ML rate: $FM = MS / (MS + MM)$. (3)
Precision: $PR = 1 - FM$. (4)
Recall: $RC = MM / (SM + MM)$. (5)
Single-layer Confidence: $CoS = 1 - SS / (SS + SM)$. (6)
Net Gain of Accuracy: $NGA = MM - MS$. (7)
These parameters facilitate the reporting and discussion of the results and the comparisons with other algorithms.
The training results in Table 3 include the confusion matrices for all eight categories with FC in bold along with the
number of CC SL and ML pixels and their sum. During the day, FC is 84.1% for CM4 ice clouds over SF areas, with the
fraction of ML correctly identified, i.e., RC = 49%. The classification during the day is a bit better for CM4 liquid clouds: FC
= 88.7% and RC = 63%. The real risk for ice clouds is 15.9% compared to 11.3% for the liquid clouds. At night the results are
similar, although a little worse for ice clouds, with FC = 81.3%. However, the ice clouds yield a larger fraction, PR = 55%, of
true ML pixels than during the day. Fewer ML clouds are found for liquid clouds at night. More of the ML clouds are classified
as ice because the nocturnal cloud temperature retrieval is based strictly on infrared radiances (Yost et al. 2021, 2023).
Nevertheless, FC is the same for both times of day for liquid phase clouds. At night, RR for ice clouds increases to 18.7% and
drops slightly for water clouds. A total of ~12 million pixels was used in the SF training.
As suggested by Fig. 3, the efficacies of the MLANN for detecting ML clouds over snow-covered areas are considerably
reduced relative to their snow-free counterparts. While the FC values are actually greater than those during the day (Table 3),
recall drops to 35% and 45% for ice and liquid clouds, respectively, during the day. The fraction detected, CoM ~ 22%, is



even lower at night. Nevertheless, because fewer pixels qualify as ML clouds over SC surfaces, according to the definition
used here, the MLANN RR values are smaller than those over SF surfaces. It is notable that, for both SF and SC surfaces, the
ML false alarm rate is less than 55%.

**4.2 Independent evaluation**

The results from the training are encouraging, but they are not based on an independent dataset. To evaluate the robustness of
the MLANN, all 2009 Aqua MODIS data were processed with the trained algorithm. In general, the statistics for the eight
categories are all very similar to those in Table 3. To summarize the effectiveness of the MLANN, the 2009 ice and liquid ML
results for the SF/SC and day/night categories are combined in Table 4. Over SF surfaces, FC is 87.0% and 85.6% for day and
night, respectively, while the corresponding values over SC surfaces are 89.3% and 88.7%. Despite the large FC values, the
MLANN underestimates the ML fraction over SF surfaces by 5.8% and 4.8% during night and day, respectively, for the
matched CC and MODIS cloudy pixels. A total of 80 million SF pixels were processed, compared to 26 million for SC surfaces.
Over SF areas, RR is 13% for day and 15% at night. Real risks drop to 11% for SC regions. It should be noted that the ML
fractions reported here are for the number of multi-layer MODIS pixels divided by the total number of  cloudy matched CC
and MODIS pixels. Since the CERES MODIS mean cloud fraction is ~0.66, the actual fraction of MODIS pixels that are
classified as ML would need be multiplied by 0.66.
The net gain of accuracy relative to the SL assumption is an important parameter to consider in any ML detection scheme.
Using the SL assumption in cloud retrievals, FC = SS + SM. Introducing a multilayer detection method yields both false and
true ML pixels. Thus, a new source of error comes with the additional information. The net gain of accuracy is not simply
equal to FC - SS, it must account for the newly introduced error, represented in MS. The falsely detected ML clouds are a
potentially worse source of error than the SL assumption for ML clouds because in retrieving ML properties, an additional
cloud layer has to be created to satisfy the ML constraint. Thus, including a ML detection algorithm in a retrieval may not be
reasonable if FM is too large. Based on Table 3, the MLANN NGA = 7.6% and 7.3% during day and night, respectively, over
SF surfaces. The corresponding values over snow-covered surfaces are 3.4% and 1.0%. While the MLANN provides a nearly
negligible amount of information over SC at night, elsewhere it clearly represents an improvement over simply assuming that
all clouds are single-layered.
Global distributions of the mean 2009 ML cloud fraction from the validation results are plotted in Fig. 4. Compared to the
average CC ML fractions (Fig. 4a), the daytime MLANN means (Fig. 4b) are distributed similarly, but are noticeably lower.
The fractions are smaller in nearly every location. At night (bottom), the patterns are much like those during the daytime,
except in the polar regions. More CC pixels (Fig. 4c) are classified as ML in the tropics than during the daytime. A comparable
increase occurs in the MLANN nocturnal results (Fig. 4d), which have ML fractions over much of the Amazon Basin and
central Africa that are comparable to their CC counterparts, although they are smaller elsewhere. As expected from Table 2,
the MLANN ML fractions in the polar regions are relatively small during the day and negligible at night.



The latitudinal variations of the mean ML fractions are plotted in Fig. 5. As expected from Fig. 4, the zonal patterns
are much the same with the MLANN values (triangles) being consistently less than their CC counterparts (circles). In the
tropics, the daytime differences generally fall between -0.06 to -0.04 and drop to as low as -0.08 in the polar regions. During
the night, the minimum of -0.10 is found over the polar regions, but the differences are comparable to the daytime values
between 45°S and 45°N. The MLANN is clearly less effective during the night over snow and ice-covered areas, especially at
night. Overall, the MLANN underestimates the 2009 ML cloud amount by 0.05 and 0.06 relative to the CCML cloud fraction
during day and night, respectively.

## 5 Discussion

These results represent a significant improvement over the previous MLANN formulation. Much of the increased accuracy is
due to the shorter horizontal averaging distances used here. By employing CALIPSO averages over distances up to 80 km,
Minnis et al. (2019) attempted to detect ML cloud systems that included many cirrus clouds having optical depths smaller than
0.20. Such clouds are difficult to detect with passive remote sensing even when they are single-layered. According to Yost et
al. (2021), systems having $\tau_{CC} < 0.2$ account for ~42% of all ML clouds for CALIPSO data using HA $\leq$ 80 km compared to
only 18% for HA $\leq$ 5 km during the day. Similar differences were observed at night. By using the smaller averaging distance,
the fraction of available ML clouds for this study dropped by almost 25% relative to that used in Minnis et al. (2019). Other
sources for the improvement arise from using additional input parameters, including those based on the 13.3-μm channel and
the reflectances at 1.38, 1.61, and 2.13 μm, as well as the vertical profiles of relative humidity. Additionally, the assumption
that all pixels having $\tau_{CM} < 0.5$ are automatically SL, regardless of the CALIPSO classification, probably removed some
difficult but less important cases.

### 5.1. Dependence on cloud properties

Much like other retrievals, the MLANN is sensitive to various cloud conditions, such as the altitudes of the two layers and
their respective optical depths. Because the MLANN uses a minimum separation distance of 1 km between the ice and liquid
cloud layers, the dependence on separation distance is not explicitly considered here. Its impact on MLANN, examined by
Sun-Mack et al. (2017) and Minnis et al. (2019), is similar to that from other studies. Tan et al. (2022), for example, found that
the probability of ML detection using a random forest method was greatest for separation differences of 3 km or more and that
it dropped from values exceeding 0.8 to less than 0.5 for cloud gaps smaller than 1 km. Greater discrepancies in altitude
between the upper and lower clouds increase the differences in the layer temperatures yielding stronger signals in the thermal
channels. It is assumed that this type of dependency, found in the aforementioned MLANN studies, is operative for this version
of the MLANN. Despite the apparent increase in accuracy using wider separation in the training, Minnis et al. (2019) found
that NGA was 60% greater for 1-km separation compared to the 3-km separation dataset. Moreover, the smaller separation
yielded nearly 50% more true ML clouds than the greater separation. The increase in apparent accuracy in the dataset using a





minimum 3-km gap relative to its 1-km counterpart is primarily due to assuming that a significant fraction of the ice-over-
water systems is single-layered, even though there is separation and two different phases in the column.
As formulated here, the MLANN assumes that all clouds with $\tau_{CC} < 0.5$ are SL. To examine this assumption, the MLANN
was also trained without any minimum COD limit. On average, FC dropped by 1.2% and the total fraction of ML clouds from
CC increased by 1.8%. Despite the drop in FC, NGA rose by 0.1%. Thus, the net impact is small and the downstream task of
unscrambling the upper and lower cloud properties from a cloud system with such a small COD will be eased somewhat.
For the radiation budget, some of the most important factors are the CODs associated with the detected and missed ML
systems. To determine the efficacy of the MLANN as a function of COD, the MLANN recall is plotted in Fig. 6 for each ($\tau_{CC}$,
$\tau_{CM}$) bin for 2009. In the plots, $\tau_{CC}$ is the ice COD for ML clouds, i.e., the upper layer COD. Irregular scales are used for the
axes to provide more detail for the lower COD values. Because of large uncertainties and reduced sampling, bins having $\tau_{CC}$
> 20 are not reliable. During the day, RC is greatest (~90%) for the bins having $\tau_{CC} \sim 1.7$ and $\tau_{CM} \sim 11$ for both ice (Fig. 6a) and
water clouds (Fig. 6b). RC exceeds 0.5 for ice clouds having $\tau_{CM} > 3$ and $0.3 < \tau_{CC} < 5$. When $1.5 < \tau_{CM} < 3$ and $0.3 < \tau_{CC} <$
1.3, RC remains above the halfway mark. The shape of the 50th percentile envelope for the water clouds differs from the ice
clouds as a result of more upper-cloud CODs being smaller than for ML clouds identified as ice (Yost et al. 2021). Thus, the
training for liquid clouds produces better ML detection when the ice clouds have small CODs.
At night, $\tau_{CM}$ is based only on thermal channels and, therefore, is mostly constrained to values of 8 or less. Default values
of 8, 16, and 32 are employed whenever the cloud is assumed to be optically thick. The particular default value depends on
the circumstances (Minnis et al., 2021). Sometimes, the CM4 and CM4+ analytical COD retrievals produce a value exceeding
8. Typically, $\tau_{CM}$ is closer to the upper-cloud COD at night, being influenced little by the lower cloud when the separation
distance is large. Ignoring the high $\tau_{CC}$ bins, the nocturnal RC maxima are found around bins (1.1, 2.0) and (1.3, 4.5) for ice
(Fig. 6c) and water clouds (Fig. 6d), respectively. True ML clouds are found more often than false SL clouds for $1 < \tau_{CM} < 5$,
when the phase is ice and $0.1 < \tau_{CM} < 3$. The halfway COD bounds narrow to $0.2 < \tau_{CC} < 0.4$ for greater values of $\tau_{CM}$. For
water-phase clouds, RC > 50% occurs mostly for $1.5 < \tau_{CM} < 8$ and $1.5 < \tau_{CC} < 4$. It is clear that the thermal channel method
is sensitive to thinner upper clouds compared to the daytime methods when the solar channel signal is overwhelmed by the
lower cloud reflectances. Conversely, the daytime method detects more ML clouds when $\tau_{CC} > 3$ or so.
This is more evident in Fig. 7, which shows histograms of the matrix parameters as a function of upper-layer $\tau_{CC}$ for SM
and MM and $\tau_{CM}$ for SS and MS over SF surfaces. For water phase clouds (Fig. 7a) the relative frequency of true ML pixels
(MM), shown as solid lines, is greater at night than during the day when $\tau_{CC} < 0.5$, but the occurrence of daytime MM pixels
exceeds their nighttime counterparts when $\tau_{CC} > 0.9$. Similar behavior is seen for the ice phase pixels (Fig. 7b), but the
thresholds shift from 0.5 to 1.4 and from 0.9 to 1.9. The false SL or missed ML (SM) clouds, shown as dashed lines, vary
differently. For the ice phase pixels (Fig. 7b), the SM pixel frequency rises with increasing $\tau_{CC}$ up to ~8% at $\tau_{CC} = 3.5$ before





decreasing to 5-7% , then dropping toward zero at $\tau_{CC} = 25$. This peak for the thick ice clouds reflects the difficulty of inferring
a lower layer under a nearly opaque cloud. For water-phase clouds (Fig. 7a), SM is most common for $\tau_{CC} < 0.3$ and diminishes
steadily to near zero around $\tau_{CC} = 30$. As $\tau_{CC}$ increases, the ML system is more likely to be identified as ice phase, so fewer
cases of ML systems having large upper-layer COD will be included in this population. In both cases, the night and day SM
frequencies track each other relatively closely with $\tau_{CC}$. Similar variations are found over the SC surfaces (not shown).
Cumulative probability distribution functions  based on the SF and SC results, presented in Fig. S1, show that 50% of the
missed ML clouds have $\tau_{CC} < 0.25$ for SF water clouds and $< 0.5$ for SC water clouds.
Figures 7c and 7d show the frequency histograms of SS and MS for CC SL liquid and ice clouds, respectively, as a
function of $\tau_{CM}$. As expected, the peak SS (true SL) frequency occurs for $\tau_{CM} < 0.5$ for both phases, day and night. During the
day, a secondary true SL maximum is found around $\tau_{CM} \approx 25$ for ice and water clouds. At night, that secondary peak is around
$\tau_{CM} \approx 14$ for ice pixels and near $\tau_{CM} \approx 9$ for liquid clouds. Nocturnal false ML (MS) clouds are found mostly between CODs
of 1 and 5 at night for ice pixels and between 2 and 6 for water clouds. During the day, MS occurs most often for $\tau_{CM} \approx 14$ for
water clouds. In fact, the MS frequency seems to follow the SS values, except for $\tau_{CM} < 0.5$. The daytime MS occurrence is
relatively flat for ice clouds with $\tau_{CM} > 1.0$.
Another factor that can influence ML detection is the assumption that the lower cloud layer is composed of liquid water
whenever the cloud temperature is less than 253 K. While that is true for most clouds, a small fraction of ice clouds have top
temperatures above 253 K (e.g., Hu et al., 2010). In those instances, the ML signal would likely be reduced because of
similarities in the optical properties of the two layers. Mixed-phase clouds, which often occur in the supercooled temperature
range, would have a similar effect, but to a smaller degree depending on the amount of ice in the cloud. On the other hand,
supercooled clouds globally account for about half of the clouds having an infrared CTT between 243 K and 253 K. If only
snow and ice surfaces are considered, the range is 239 K to 242 K (see Fig. 6 of Hu et al., 2010). Thus, some systems with
cold (CTT < 233 K) ice clouds over supercooled liquid clouds with CTT < 253 K could be identified as SL ice by the definition
used here. These complementary effects due to supercooled clouds could produce some confusion in the training of the
MLANN, particularly in polar regions.
The CODs used in the training would not be the same as those determined using the standard CM4 algorithms employed
for the 2009 retrievals because the CM4+ algorithms used a different ice crystal model and a new method for retrieving COD
over snow. This change in COD retrieval apparently had minimal impact on the detection as the 2008 training and 2009
validation results are nearly identical.
**5.2 Comparisons with other results**
As noted earlier, multilayered cloud detection has been the subject of many different algorithmic studies, so it is important to
better understand how the current approach compares to those other algorithms. Direct comparisons are not straightforward



because each algorithm was developed with its own specific constraints and ML definitions. The CERES Ed4 ML algorithm
(Chang et al., 2010a, Minnis et al., 2021) was applied only when a cloud with pressure below 500 hPa was detected using a
$CO_2$-absorption method (Chang et al., 2010b). The MODIS science team algorithm (Wind et al. 2010) was applied to a 5-km
cloud product and was only used when the MODIS optical depth exceeded 4. The latest version, MYD06 C6.1 (Platnick et al.,
2017), adds the $BTD_{1112}$ technique developed by Pavolonis and Hiedinger (2004). Desmons et al. (2017) used data from the
Polarization and Directionality of the Earth's Reflectance (POLDER) to detect ML clouds of all types, but only for $\tau > 5$. Ice-
over-water multilayered clouds were detected by Wang et al. (2019) during daytime using Suomi-NPP (SNPP) Visible Infrared
Imager Radiometer Suite (VIIRS) data in a thresholding method. Tan et al. (2022) placed no restrictions on either $\tau$ or the
number of layers, but they applied their random forest algorithm and other machine learning techniques only to geostationary
Himawari-8 data. Because of its orbit, the Himawari-8 observations are taken over a full range of VZA when matched with
the CC profiles, but the VZA is constant for a given location. Other published methods have either not produced extended
datasets or performed only case-study evaluations with objective data. Despite the sampling disparities, it is informative to
compare some of the statistics to provide some context to the performance of the MLANN. These comparisons are summarized
in Table 5.

Comparing with CC data, Desmons et al. (2017) found that for overcast clouds with $\tau > 5$, FC = 70% and CoS = 74%.

For the same conditions, they determined that MYD06 C6.1 yields FC = 67% and CoS = 73%. Additional parameters computed
from their Table 4 are listed in Table 5. Precision and recall from MYD06 are 54% and 47%, respectively, while they are 58%
and 47% from POLDER. These can be compared to the MLANN daytime validation results (Table 5), which combine the SC
and SF daytime data in Table 4 weighted by 0.13 and 0.87, fractions that roughly correspond to the areal coverage of the
respective surface types (e.g., Yost et al., 2023). All of the MLANN parameter values exceed their restricted MYD06 and
POLDER counterparts. Wang et al. (2019) only reported validation results in terms of percent of CALIOP ML and SL. Thus,
only RC and CoS could be determined from their results. For $\tau > 1$, the recall is about the same as the daytime MLANN, if the
ML and probably ML categories from their algorithm are combined. Similarly, their CoS is around 10% smaller than the
MLANN value. If clouds with $\tau < 1$ are included, both CoS and RC drop substantially. Note that Wang et al. (2019) did not
include CloudSat retrievals in their evaluation, so ML clouds with an optically thick upper cloud are not included in the
statistics.

Although no value for FC was given, the values of certain parameters can be estimated for all clouds with an unrestricted

optical depth from the figures in Desmons et al. (2017). From their Fig. 8, RR ≈ 38%, so FC = 62%. The value of CoS is the
same for restricted and all clouds. The MODIS parameters change only negligibly for all clouds compared to the restricted
case because the MYD06 algorithm only uses clouds with $\tau > 4$. Marchant et al. (2020) also compared the MYD06 to the 2B-
CLDCLASS-lidar products and found that for clouds with $\tau > 4$, FC = 63% with the Pavolonis and Heidinger (2004) algorithm
and 65% without it. If it assumed that all clouds with $\tau < 4$ are SL, then FC jumps to 80% and 81% for the two algorithm
options. But that assumption excludes 45% of the ML clouds as defined by Marchant et al. (2020).



Except for the definition of what constitutes a ML cloud (ice over water, water over water, etc.), NGA is the one parameter
that is not too dependent on cloud optical depth assumptions. From Desmond et al. (2017), the daytime POLDER and MYD06
C6.1 cases yield NGA = 4.4% and 2.2%, respectively. Presumably, some of the POLDER results include water-over-water
clouds. Nevertheless, the POLDER algorithm yields a net gain in information. The results of the Marchant et al. (2017) analysis
yield slightly lower numbers for the MODIS C6.1 NGA, 0.2% and 1.4%, with and without the Pavolonis and Heidinger (2004)
method. In either case, the MLANN daytime NGA exceeds those of the MYD06 and POLDER techniques. Moreover, it greatly
exceeds the CERES Ed4 ML results (not shown).
The random forest results from Tan et al. (2022), confined to 60°S - 60°N and 80°E and 160°W, were trained with 1-km
matched 2B-CLDCLASS-LiDAR profiles using the product's layer flag to determine if a given pixel is SL or has more than
one layer, regardless of layer phase. That training dataset produced ACC = 85% and 79%, respectively, for the daytime and
all-hours algorithms. The latter method included no reflectance input from solar channels so it can be used for both day and
night conditions. It is included in the bottom section of Table 5 for comparing to the MLANN night version. For this technique,
PR = 81% and 73% for day and all-hours, respectively, with corresponding RC values of 72% and 64%. While ACC is less
than that found with the MLANN for both day and night, the random forest PR and RC results are greater than their MLANN
counterparts. At night, the MLANN PR is nearly equal to the Himawari-All value.  The Himawari CoS and NGA daytime
values were deduced from the values in their Table V and their equations (1) - (3). The MLANN CoS values exceed their
Himawari counterparts, but the MLANN global NGAs are less than half of those from the random forest training results. Those
larger values arise, in part, from including many more types of ML clouds in the random forest training than used for the
MLANN.
A fairer comparison would use independent validation sets from both algorithms, but a complete summary of the
validation comparisons was not provided in Tan et al. (2022). However, several parameters can be determined from their Fig.
5, which utilized a dataset independent of the training data. The resulting values of PR are 70% and 64% for day and all-hours,
respectively, while CoS = 89% and 85%. The MLANN SL confidences are slightly greater at 90% and 88% and its PR values
exceed the Himawari validation results, especially for night/all-hours. Without further information it is not possible to
determine the values of ACC and RC for the geostationary validation dataset. However, because CoS is the same or larger for
the validation dataset and PR dropped by 11 points from the training results, it can be inferred that the fraction of false ML
clouds increased considerably. This would reduce ACC and substantially diminish NGA.
Interestingly, the best results from the Tan et al. (2022) validation analysis are for ice-over-liquid and ice-over-mixed
clouds. The former corresponds to conditions that the MLANN was developed to detect, while at least some of the latter were
included in the MLANN analysis. Approximately 30% of the actual ML clouds detected in the Tan et al. (2022) validation
analysis are for single phase or upper-layer mixed phase ML clouds that MLANN was not designed to identify. Assuming that
the portion of the ice-over-water/mixed is the same for the training dataset, the correctly detected ice-over-water cloud amount



is 0.10. Adding the ice-over-mixed would yield 0.17. Reducing that by the ratio of PRs from the validation and training sets
would drop the range to 0.09 - 0.15, which bounds the correct ML fraction from the SF cases in Table 4.

**5.3 Full-swath detection**

The MLANN training is based on near-nadir measurements from both the CC and MODIS instruments. Increasing optical path
lengths due to increasing VZA modify the radiances emanating from a given location through absorption and scattering. This
is particularly true for radiances at solar wavelengths. Thus, the near-nadir-based MLANN coefficients are not necessarily
valid for observations taken at other VZAs. For operational use with Aqua MODIS data, the MLANN must be reliable across
all viewing angles.

**5.3.1 Angular dependence**

The VZA dependency is examined by first computing the mean radiances for each viewing angle across the full scan for data
taken during JAJO 2019. It was found that the radiance VZA dependence is sensitive to the forward or backward portion of
the scan cycle. The former view is toward the sun, while the latter is directed away from the sun. Figure 8 plots the reflectance
averages for each VZA bin in the forward (positive) and backward (negative) directions. From near-nadir to 65°, the 1.60-µm
reflectance (solid lines) for water cloud pixels increases by 11% and 25% in forward and back directions, respectively. For ice
clouds, the corresponding increases are 22% and 37%. Similar changes are seen for the 2.13 µm reflectances (dashed lines).
The 1.38-µm reflectance for ice behaves in much the same manner (Fig. S2), but is nearly constant with VZA for water clouds.
The daytime 3.75-µm radiances (Fig. S3) are relatively flat in the back direction, but increase with VZA for liquid clouds. At
night, the radiances show the classic limb-darkening behavior of thermal radiation. This can be seen in Fig. 9. During the day
(solid lines), the water-cloud 10.8-µm radiances are relatively flat in the forward direction and drop a little at the higher VZAs
in the back direction. Ice cloud radiances decrease in both directions, but more so in the forward view where the 10.8-µm
radiances are lower than their back-direction counterparts. The forward scan views more shadowed areas that could affect the
thin cloud and partly cloudy scenes over land (Minnis et al., 2004). At night (dashed lines), the limb-darkening is more
apparent. No back and forward differences are considered at night. Similar variations in radiance are seen at 8.55-µm (Fig. S5)
and 11.90-µm (Fig. S6). There are only minor radiance differences between the forward and back directions during the day
for the 6.70-µm channel (Fig. S4), presumably because it is mostly unaffected by the layers below the cloud. Additional plots
of radiances as a function of VZA (Figs. S6 -S14) are provided in the Supplemental Material.
To adjust the MODIS radiances, ice and water correction factors were determined for each waveband, day and night,
separately over SF and SC surfaces. For daytime, the correction factors were computed for both forward and back scans. These
factors were developed for both the channel radiances and reflectances and each of the BTD parameters. The correction factor
is simply the ratio of the mean radiance for VZA between -3° and -18° divided by the mean radiance at a given VZA. Thus,
the observed radiance is adjusted to the near-nadir view of MODIS by simply multiplying it by the correction factor.



To test the impact of these factors on the retrievals, the MLANN was applied to the uncorrected and corrected full-swath
MODIS data for April 2009. Figure 10 shows the variation of mean ML fraction as a function of VZA for SF ice and water
clouds, day and night. During the day (Fig. 10a), the uncorrected and corrected ML fractions are nearly identical suggesting
that the correction factors have minimal effect on the radiances. While this is not surprising, given the relatively flat daytime
curves in Fig. 9 and for other thermal channels, it is actually due to a cancelation of the ML decrease in the back direction by
an increase in the forward direction (not shown). In contrast to the daytime results, the nocturnal ML fractions have a
nonmonotonic variation with VZA for the uncorrected radiances and a significant steady decrease to a value near zero for the
corrected case. The uncorrected radiances for ice clouds (Fig. 10b) yield a rise in ML detection during the day and a drop at
night. When the correction factors are applied to the radiances, the ML amounts are relatively constant with VZA. To obtain
the most consistent product across the swath, the adjustments are applied to all of the radiances, except for water cloud pixels
observed during the night.
**5.3.2 Example images**
Figure 11 shows the results of applying the MLANN with VZA correction to an Aqua MODIS image taken over the Southern
Ocean centered near 57°S, 165°E at ~3:50 UTC, 16 April 2019. The pseudo color RGB image (Fig. 11b) shows an extensive
area of stratocumulus on the left side that is apparently overlaid with thin cirrus that blurs the view of the underlying clouds.
A second extensive liquid cloud deck appears near the top center that might overlay some StCu clouds, but is itself covered
by thicker ice clouds. The CM4 cloud phase results (Fig. 11a) highlight those contiguous dense ice clouds, which likely obscure
lower clouds. Thin cirrus also appear to overlie parts of the second liquid deck. The MLANN (Fig. 11c) determines that a large
portion of the image consists of ice-over-water clouds. In general, the outline of the cloud effective heights (CEH) above 1 km
correspond to the ML pixels (Fig. 11d) except where the ice cloud is very thick, or perhaps, the ice cloud is in close proximity
to or contiguous with the lower deck, as over parts of the white clouds in top center part of the image. Other higher, SL liquid
clouds are seen near the top left corner and bottom right of center. Cloud phase is very mixed over the thin cirrus areas, yet
the MLANN determines most of the pixels as ML.
Multilayered clouds detected with the nighttime MLANN are shown in Fig. 12 for a MODIS image taken around 4:45
UTC the same day over the North Atlantic. The scene (Fig. 12b) contains extensive but variable cirrus coverage (white) and
broken stratus clouds typically between 1 and 3 km (Fig. 12d). Thicker cirrus clouds are identified as ice (Fig. 12a) while many
of faintest ones, primarily those over the low clouds, are classified as liquid. Denser ice clouds and those over open water
appear to be at altitudes between 9 and 14 km, while the thin Ci over St range from 3 to 7 km, which is expected, given the SL
cirrus altitudes. The MLANN appears to identify many of those ML clouds (Fig. 12c), but tends to miss those with overlying
thick cirrus. There may be some false ML clouds in the upper right, but it is difficult to tell because the thinnest Ci is not
always discernible in the RGB image.



The final example shown here (Fig. 13) is taken the same day around 01:50 UTC over the polar ice cap centered near
80°N, 155°E.  Snow and ice cover provide the scarlet background (Fig. 13b), which is overlaid with low clouds in various
shades of white to gray and a slightly higher deck in the center with thin Ci covering much of the top half of the image. That
cirrus appears as blurry pinkish gray and identified as ice or liquid depending on the thickness (Fig. 13a), while most of the
cirrus over the deck in the middle is designated as liquid. Those clouds are identified as ML by the MLANN along with the
small area at the bottom and in the upper left (Fig. 13c). Only a few parts of the cloud left of center are classified as ML, while
it appears more ML clouds should have been detected.  Most of the SL ice cloud CEHs are only between 3 and 6 km (Fig.
13d), while those over the middle deck are less than 3 km. The low CEH values are likely due to overestimation of the COD
by the CM4 retrieval for SL clouds and to the presence of the thick lower cloud for the ML retrievals. Detection of the ML
clouds will allow reclassification of the cloud tops as ice and recalculation of the cloud properties, when the components of a
two-layer retrieval system are in place.
These three examples and the two additional cases shown in Figs. S15 and S16 demonstrate that MLANN performs
reasonably well across the full swath. No wild false ML clouds are evident although some recognizable misses are seen, as
expected from the analyses above. Quantifying the accuracy of the correction-factor approach to full swath application of the
MLANN would ultimately require using the method on similar data taken by a different satellite, such as SNPP VIIRS, that
overlapped with the CC data at various VZAs. Using VIIRS data, Wang et al. (2019) found that the ML clouds they detected
showed minimal changes with VZA. That result is similar to the daytime curves in Fig. 10b. Developing and analyzing a
comparable VIIRS-CC dataset is beyond the scope of this paper, but is planned for future research.

### 5.3.3  Assessment of full swath results

In the interim, more indirect approaches are available. For example, the off-nadir and near-nadir results should be spatially
consistent if the swath approach is working properly. To examine this aspect, Figure 14 shows the distributions of ML cloud
fractions averaged over the months of January, April, July, and October (JAJO) 2009 from three different data sources. These
include daytime retrievals from all CC data (Fig. 14a), MLANN applied to Aqua MODIS radiances observed at the reference
near-nadir (-3° < VZA < 18°) angles (Fig. 14b), and to Aqua MODIS data taken at all VZAs (Fig. 14c). The corresponding
nocturnal results are plotted in Figs. 14d-f. These results are noisier than those in Fig. 4 because they are based on only 4
months of data and they include all observations, not just those having good CC and C3M cloudy pixel matches. The MODIS
results include both false and partially cloudy pixels.
As in Fig. 4, the CC and near-nadir patterns are comparable although the MLANN means are often smaller than their CC
counterparts. The areas with minimal ML amount in the near-nadir results (Fig. 14b) are in the same locations as those from
the CC retrievals, but are more pronounced. Some CC maxima are reproduced by the MLANN, but the MLANN fractions
near the maxima drop off more precipitously than their CC counterparts. For example, the maximum off the southern Chilean
coast in Fig. 14b is nearly identical that in Fig. 14a, but the MLANN fractions in the surrounding areas are generally smaller



than the CC values. While much smoother, the non-polar patterns in both all-VZA cases (Fig. 14c) are similar to those from the near-nadir results, but the Chilean maximum is diminished somewhat.

Distributions of ML fractions from the same datasets are even more consistent at night. The maxima over northern South America, central Africa, and Indonesia are well defined in all three maps. Like the daytime results, the non-polar minima are much better delineated in Figs. 14e and 14f than in the CC data (Fig. 14d). Overall, the distributions in Fig. 16 demonstrate that the full-swath MLANN does not yield spurious ML clouds in areas where they are not expected to occur and generally produce results similar to the near-nadir values.

Another measure of robustness of the algorithm is its ability to reproduce the seasonal cycle. This is examined by computing the monthly mean ML anomaly, which is defined as the monthly mean minus the annual average divided by the annual average. It is clear that the SC results over snow miss many ML clouds, especially at night. Thus, to minimize the influence of SC regions on the seasonal cycle, only nonpolar (60°S - 60°N) data are considered. Figure 15 plots the ML fraction anomaly for each month of 2009 from CC and the MLANN applied to full-swath Aqua MODIS data. The MLANN day and night anomalies track their CC counterparts remarkably well, within a few percent in most cases.

To further examine the reliability of the MLANN on longer time scales, it was applied to January, April, July, and October (JAJO) 2013 Aqua MODIS full swath data. The global distributions of the 2009 and 2013 results (Fig. S16) are similar, but reveal shifts in the locations of the maxima. Table 6 presents the global mean JAJO 2009 and 2013 ML fractions along with the land-ocean ratio, L/O, which is the global average ML fraction over land divided by that over water surfaces. Overall, ML fractions for all CC data are 3 - 5% greater than their MODIS-matched counterparts, a result comparable to the differences in Fig. 4. The 2009 MLANN near-nadir values are 0.01 smaller than those for all VZAs. ML fractions in Table 6 are all less than their counterparts in Fig. 4. This is due to the fact that the CC data in Table 6 include all cloudy pixels that the CERES cloud mask classified as clear and the MLANN results include many partly cloudy pixels that are not likely to be ML. The clouds detected by CALIPSO, but missed by CERES are mostly SL thin cirrus and SL low clouds (Yost et al. 2021, 2023), which would dilute the ML fraction determined using all of the CC data. The differences between the CC and near-nadir MLANN are reduced by ~2% compared to those using only the matched data. During daytime, the MLANN mean ML fractions from 2013 are 0.5% greater than those in 2009, while at night the 2013 averages exceed their 2009 counterparts by 0.2% near nadir and 0.6% across the full swath. For both years, the nocturnal near-nadir values are ~1% less than for data taken at all VZAs.

The CC land-ocean ratios, L/O, in Table 6 reveal that fewer ML clouds occur over land than over water surfaces. For CC, L/O is between 0.77 and 0.84, while it varies from 0.64 to 0.77 for the MLANN results, indicating that the MLANN is less efficient at detecting ML clouds over land than over water bodies. Together with the similarity of the CC and ML seasonal cycles, the consistency of the near-nadir and full-swath L/O values and small differences in ML amounts during both years are quite encouraging for using the MLANN on an operational basis.



### 5.3.4 Operational considerations

CERES is a long-term project that utilizes many different satellites and imagers to characterize cloud properties. The MODIS
on Aqua and Terra and the VIIRS on SNPP and NOAA-20 are coincident with the CERES broadband radiometers and observe
non-polar regions at fixed times each day. Any system designed to detect ML clouds should be applicable to both the VIIRS
and MODIS imagers and, ideally, to the geostationary imagers that are used to help assess the radiation budget at other times
of day. Because the latter have had widely varying spectral channel complements since 2000, use of MLANN with them is
beyond the scope of this discussion.  The VIIRS lacks certain channels used here (13.3 μm and 6.7 μm) and the channels
common to MODIS and VIIRS differ in spectral coverage and filtering. Additionally, the VIIRS is a higher resolution
instrument. Thus, it may be necessary to train VIIRS with CC data to obtain a consistent ML result. Another approach would
require careful inter calibration of the VIIRS and MODIS channels using spectral corrections (e.g., Scarino et al., 2016) and
the addition of radiances from the missing channels determined from a process that fuses data from VIIRS and the Crosstrack
Infrared Sounder (e.g., Weisz et al., 2017). If those are not available, then the MLANN would need to be retrained with fewer
input radiances, likely at the expense of accuracy. To that end, initial training tests indicate that without those channels, ACC
decreases 87.0% to 86.4% during the day and from 85.6% to 84.3% during the night over SF surfaces. During the day, NGA
drops from 7.6 to 7.1%, while at night NGA goes from 7.3 % to 6.2%. NGA is relatively unaffected by the loss of the 13.3-
μm channel; almost all of the diminished accuracy is due to the absence of the 6.7-μm channel, particularly at night. Even with
the loss of those channels, the resulting detection capability would still represent a significant advancement over previous
efforts.
As in all retrievals, reliable and consistent calibration across platforms is essential to providing an accurate ML product.
It may be even more important for the MLANN because the neural network relies on subtle radiance differences that may be
lost in the noise of a physical retrieval. Thus, any small trend in the calibration of one channel may introduce a growing bias
in the ML fraction. Similarly, inter platform calibration differences could cause a similar effect. Updated retrieval algorithms
and input data are introduced into the CERES data processing whenever major improvements are developed and errors
diminished. Since the MLANN relies on a few retrieval inputs such as COD and cloud phase, it would need to be retrained
whenever a new CERES cloud algorithm edition is introduced.
Further improvement of the MLANN itself, particularly over snow-covered areas, might be gained by using additional
parameters or spatial context. For example, Tan et al. (2022) found that radiances from the 7.3-μm channel comprise a highly
ranked predictor of ML clouds in their random forest approach. The MODIS equivalent channel was not considered here, but
would have to be created for VIIRS using the fusion process noted above. Information about the pixels surrounding the pixel
of interest increased the accuracy of ice water path retrieved from a Meteosat imager with a convolutional neural network
(Amell et al., 2022). Including selected radiances or BTDs from surrounding pixels might also enhance the MLANN.
Additional partitioning of the training categories might also raise ML detectability as it did when the original MLANN (Sun-



Mack et al, 2017) was divided into ice and water phase categories (Minnis et al. 2019). These and other approaches could lead
to greater accuracies than found here.

## 6. Summary and Conclusions

An artificial neural network method has been enhanced to more accurately identify ice-over-water ML cloud systems from
multispectral MODIS observations. The algorithm requires as input a variety of radiances, brightness temperature differences,
atmospheric profiles of temperature and humidity, and the CERES Edition 4 cloud phase and optical depths. Based on the
definitions of single and multilayer clouds used here for CALIPSO-CloudSat profiles, the MLANN correctly identifies SL and
ML clouds together 87.0% and 85.6% of the time over surfaces free of ice and snow during day and night, respectively. Over
ice or snow-covered areas, the corresponding correct identifications are 89.3% and 88.7%. Despite the good overall agreement,
the MLANN only detects 55% of the CC ML clouds over SF regions and only 40% and 20% over SC areas during day and
night, respectively. The majority of the missed SF ML clouds are those having an upper-cloud COD < 0.3 (water) or COD $\geq$
3 (ice), although ~35% and 20% of the water and ice-phase detected ML clouds meet those conditions. Over SC surfaces, the
undetected ML pixels mainly have an upper-cloud COD < 0.5 or COD > 2.
Despite its shortcomings, the MLANN, unlike many other techniques, yields a significant net gain in layering
identification accuracy because the number of false ML pixels is substantially less than that for true ML pixels. Overall, the
daytime MLANN evaluation metrics are more favorable than those based on physical retrievals or decision tree algorithms,
even with the differences in sampling, ML cloud definitions, and optical depth constraints. Few methods have been developed
for nocturnal application. Comparisons with results from a machine learning algorithm applied to geostationary satellite data
have yielded a more ambiguous assessment. The accuracy and SL confidence from MLANN are greater than those from the
Tan et al. (2022) random forest training results for day and night. Yet, the MLANN precision, recall, and NGA values are
smaller. If the validation results from Tan et al. (2022) are considered, the MLANN precision values are greater. It is not
known how much the MLANN recall and NGA would fare relative to the random forest validation results. Even if it were
known, the relative merits of the two methods would be difficult to quantify without accounting for the discrepancies in ML
definition and sampling areas and time periods. However, it can be concluded from the comparisons that the MLANN is among
the most capable of current ML detection methods.
Operationally, the MLANN, trained with near-nadir MODIS views, must be applicable to all the MODIS viewing angles.
To account for the variation of radiances with viewing zenith angle, the MODIS-based input parameters are normalized to the
nadir view using empirical correction factors. The adjustments yield ML cloud amounts that are mostly invariant with VZA
and produce visually reasonable ML detection across the MODIS swath. Spatial distributions of ML cloud fractions from full-
swath results are consistent with the near-nadir results and manifest similar detection efficiencies over land and water surfaces
that are the same as their near-nadir counterparts. Temporally, the MLANN produces the same seasonal cycle in ML clouds



as the active sensor data, albeit with the noted bias. Moreover, the results are similar in magnitude and distribution for different years with shifts in maxima. While more detailed pixel-to-pixel comparisons should be performed using CC data matched to imagery taken at off-nadir VZAs, the analyses performed here indicate that the MLANN should be as successful off of nadir as it is in the near-nadir mode.

Applying the MLANN to other imagers should be performed with caution as sensors on other satellites can differ spectrally and spatially (e.g., VIIRS) or may observe at other times of day (e.g., Terra MODIS). Platforms that are not in Sun-Synchronous orbits, for example geostationary satellites, will observe a given scene at times of day and at viewing and illumination angles that are not seen by Aqua MODIS and hence not in the training complement. Adapting the MLANN to different types of orbits or times of day presents a challenge as there are few options for global training and validation. Current and future cloud radar and lidar combinations are confined to afternoon Sun-synchronous satellites (e.g., Heliere et al. 2017). Lidars that can be used for cloud detection have flown on the International Space Station (e.g., Pauly et al., 2019) in a precessing orbit and on Aeolus in a sunrise/sunset Sun-synchronous orbit (Straume et al., 2020). CALIPSO has been slowly moving away from its 1330 LT orbit covering several more hours of the day since 2018. Without the cloud radar, any and all of those lidars could be used to define ML clouds to some extent, depending on their penetration depths, and may be of value for training and validating ML clouds for geostationary imager data. Regardless of the particular satellite, the MLANN would need to be retrained or the spectral channels normalized to MODIS.

With layer detection accuracies below 90%, there is clearly room for future improvement, especially over polar regions covered with snow and ice. Use of additional channels or subsets of the current training categories may add a few more points to the overall accuracy. Combining physical retrievals with the neural network may also be the means for detecting more ML pixels. The definition of ML clouds used here is rather restrictive in that it is nominally confined to ice over liquid water clouds. It is also somewhat ambiguous because 253 K serves as the threshold between ice and water clouds for the underlying cloud deck. In lieu of any better information to define the lower cloud phase, the threshold should be altered to account for variability of the 50th percentile ice phase in the supercooled temperature range. Other cloud combinations such as liquid over liquid could be included in the MLANN but they might reduce the accuracy and would probably be more resolvable if treated separately from the ice over water clouds.

Detecting multilayer clouds is a first step toward improving the characterizations of global vertical cloud structure using passive sensors. Once identified, the upper and lower layer cloud properties need to be estimated. A number of approaches have been suggested for estimating the top heights of the upper and lower clouds. These include physical retrievals (e.g., Chang et al. 2010) and machine learning (e.g., Minnis et al. 2019). Similarly cloud optical depth and particle effective size could be derived with a physical retrieval (e.g., Chang et al. 2010), a neural network (e.g., Cerdeña et al. 2007), or an optimal estimation method that requires the cloud heights (e.g., Sourdevall et al., 2016). Having a reliable detection method, like the MLANN, should serve as motivation for formulating a robust technique for unscrambling the upper and lower cloud layer properties in future research.




*Data availability.*

MLANN training data used in the paper can be obtained from CERES Ordering Tool: *https://ceres.larc.nasa.gov/data/*


*Author contributions.* S. Sun-Mack, P. Minnis, and G. Hong developed the detection method. S. Sun-Mack implemented the

technique. Y. Chen, S. Sun-Mack, and P. Minnis performed the data analyses. The paper was first drafted by P. Minnis and

revised by S. Sun-Mack and W. Smith, Jr. The project was supervised by P. Minnis and W. Smith, Jr.


*Competing interests.* The authors declare no competing interests.




*Acknowledgments.* This research is supported by the NASA CERES Project.

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

**Tables**




**Table 1.** Input parameters for MLANN.

| Regional Parameters | GEOS-5.4 | MODIS Thermal Data (K) | MODIS Solar Data[†] |
|---|---|---|---|
| Latitude (°), Longitude (°) | Surface skin temp (K) | $T_{37}$, $T_{67}$, $T_{85}$, $T_{11}$, $T_{12}$, $T_{13}$[@] | $R_{CM}$, $\tau_{CM}$ |
| Surface type, elevation | Relative humidity (%) at 8 levels* | $BTD_{3711}$, $BTD_{6711}$, $BTD_{8511}$, $BTD_{1112}$, $BTD_{1113}$[@] | $\rho_{1.38}$, $\rho_{1.61}$, $\rho_{2.13}$ |
| Solar zenith angle (°)[†] | Temperature (K) at 8 levels | | $\rho_{1.61}$ - $\rho_{2.13}$ |



* levels: surface, 850, 700, 500, 400, 300, 200, 100 hPa
[†] day only
[@], snow/ice free only





**Table 2.** Confusion matrix definition.

|  | CC Single | CC Multi | Total |
| --- | :---: | :---: | :---: |
| MLANN Single | SS | SM | SS + SM |
| MLANN Multi | MS | MM | MS + MM |
| Total | SS + MS | SM + MM | SS + MM |





Table 3. Confusion matrices (each bounded by dashed lines) for MLANN applied to Aqua MODIS relative to layer identification from CloudSat-CALIPSO, 2008, from the training set. The bold numbers indicate the percent correct for each matrix.

| | **CloudSat and CALIPSO** | | | | | | | | | | | |
|---|---|---|---|---|---|---|---|---|---|---|---|---|
| | Snow-free, Day | | | Snow-free, Night | | | Snow-cover, Day * | | | Snow-cover, Night* | | |
| **MLANN** | SL | ML | Total | SL | ML | Total | SL | ML | Total | SL | ML | Total |
| Ice SL,% | 73.2 | 11.3 | 84.5 | 65.9 | 12.7 | 78.6 | 86.6 | 7.8 | 69.5 | 86.4 | 9.4 | 95.8 |
| Ice ML,% | 4.6 | 10.9 | 15.5 | 6.0 | 15.4 | 21.4 | 1.8 | 3.9 | 30.5 | 1.5 | 2.7 | 4.2 |
| Total, % | 77.8 | 22.2 | **84.1** | 71.9 | 28.1 | **81.3** | 88.4 | 11.7 | **90.4** | 87.9 | 12.1 | **89.1** |
| # pixels x $10^3$ | 3,748 | 1,070 | 4,818 | 4,097 | 1,599 | 5,696 | 2,549 | 390 | 2,884 | 11,41 | 1,500 | 12,91 |
| Liquid SL, % | 76.3 | 7.4 | 83.7 | 79.5 | 7.9 | 87.4 | 82.7 | 8.2 | 71.3 | 87.3 | 9.0 | 96.3 |
| Liquid ML, % | 3.9 | 12.4 | 16.3 | 3.3 | 9.3 | 12.6 | 2.5 | 6.6 | 28.7 | 1.3 | 2.4 | 3.7 |
| Total, % | 80.2 | 19.8 | **88.7** | 82.8 | 17.2 | **88.8** | 85.2 | 14.8 | **89.3** | 88.6 | 11.4 | **89.7** |
| # pixels x $10^3$ | 4,502 | 1,112 | 5,614 | 5,297 | 1,100 | 6,397 | 5,647 | 996 | 6,643 | 844 | 319 | 3,941 |








**Table 4.** Same as Table 3, but for combined liquid and ice results from applying MLANN to the 2009 validation dataset.

| | **CloudSat and CALIPSO** | | | | | | | | | | | |
|---|---|---|---|---|---|---|---|---|---|---|---|---|
| | Snow-free, Day | | | Snow-free, Night | | | Snow-cover, Day | | | Snow-cover, Night | | |
| **MLANN** | SL | ML | Total | SL | ML | Total | SL | ML | Total | SL | ML | Total |
| SL,% | 75.3 | 8.9 | 84.2 | 74.0 | 10.1 | 84.1 | 83.6 | 8.4 | 92.0 | 86.2 | 9.8 | 96.0 |
| ML,% | 4.1 | 11.7 | 15.8 | 4.3 | 11.6 | 15.9 | 2.3 | 5.7 | 8.0 | 1.5 | 2.5 | 4.0 |
| Total, % | 79.4 | 20.6 | **87.0** | 77.3 | 21.7 | **85.6** | 85.9 | 14.1 | **89.3** | 87.7 | 12.3 | **88.7** |
| # pixels x $10^3$ | 28,883 | 7,493 | 36,376 | 33,739 | 9,908 | 43,647 | 8,235 | 1,352 | 9,587 | 14,277 | 2,002 | 16,279 |








| Algorithm | FC/ACC | PR | RC | CoS | NGA |
|---|---|---|---|---|---|
| MYD06 C6.1, τ > 5 | 67 | 54 | 46 | 73 | 2.2 |
| POLDER 95, τ > 5 | 70 | 58 | 47 | 74 | 4.4 |
| VIIRS, τ > 1 (τ < 1) | - | 65 (53) | 65 (53) | 79 (64) | - |
| Himawari Training Day | 85 | **81** | **72** | 87 | **18.3** |
| Himawari Validation Day | - | *70* | - | *89* | - |
| MLANN day | **87** | 74 | 55 | **90** | 7.6 |
| Himawari Training All | 79 | **73** | **64** | 82 | **14.3** |
| Himawari Validation All | - | *64* | - | *85* | - |
| MLANN Night | **86** | 72 | 52 | **88** | 6.5 |

**Table 5.** Confusion matrix metrics in % for various multilayer algorithms. MYD06 and POLDER 95 are based on Table 4 of Desmons et al. (2017). Himawari results are based on Tan et al. (2022) random forest results. MLANN results based on the 2009 validation parameters in Table 4. Bold numbers denote the largest value of a given parameter for each section (separated by dotted line).







<p align="center">**Table 6.** Average ML fractions from CC and Aqua MODIS MLANN for JAJO.</p>

| Time | CC 2009 | MLANN near-nadir, 2009 | MLANN all VZA, 2009 | MLANN near-nadir, 2013 | MLANN all VZA, 2013 |
|---|---|---|---|---|---|
| Day, ML (%) | 15.4 | 12.1 | 11.8 | 12.6 | 12.3 |
| Night, ML (%) | 17.7 | 12.6 | 13.5 | 12.8 | 14.1 |
| Day, L/O | 0.77 | 0.64 | 0.65 | 0.63 | 0.64 |
| Night, L/O | 0.84 | 0.74 | 0.75 | 0.76 | 0.77 |



**Figures**



 

**Input Layer**

*See Table 1*

**Hidden Layer**

**Output Layer**

*Probability of ML*

$X_1$ Input #1

$X_2$ Input #2

$X_3$ Input #3

$X_4$ Input #4

$f_1$  $f_1$  $f_1$  $f_1$  $f_1$

Output **Y**

$W_{ij}, b_j$

$W_j, c$

**INPUT**

**OUTPUT**

$$g_j = \sum_{i=1}^{n_1} w_{ij} x_i + b_j$$

$$u_j = f_1(g_j) = \frac{2}{1+e^{-2g_j}} - 1$$

$$y = \sum_{j=1}^{n_2} w_j u_j + c$$


**Figure 1.** Schematic of artificial neural network used here.







**Figure 2.** CALIPSO-CloudSat classification frequency of occurrence for matched 2008 CERES-MODIS cloud phase selection, ice (left) and liquid (right) for snow-free surfaces (top) and snow/ice-covered surfaces (bottom).



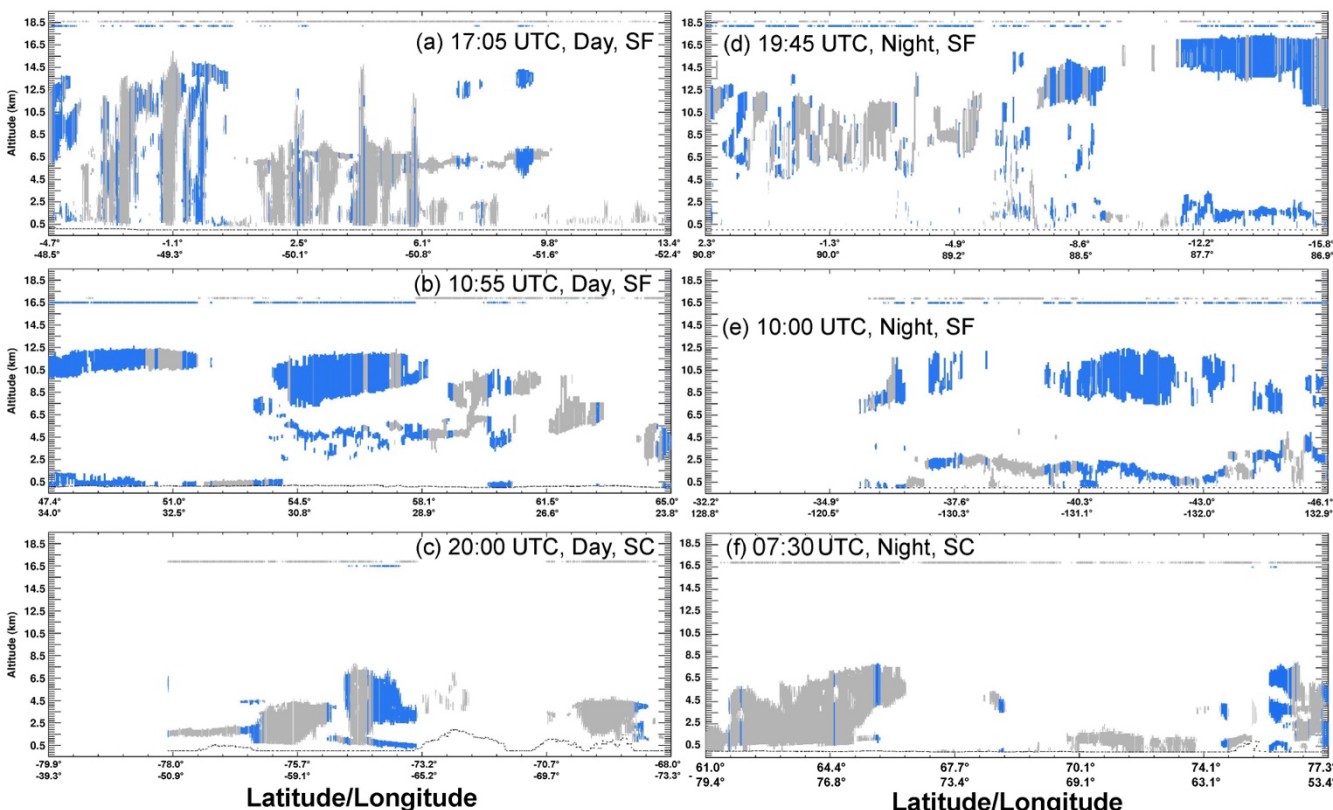

**Figure 3.** CALIPSO-CloudSat cloud profiles from C3M for 25 December 2009 with CC ML clouds indicated in blue and CC SL

denoted in gray. The MLANN ML identification for each profile is indicated as a blue dot at the top of each figure. MLANN SL

clouds are indicated with a gray dot. Surface elevation is given as the dotted line at the bottom of each panel. Tropical,

midlatitude, and polar cloud profiles are given in the top, middle, and bottom profiles, respectively.



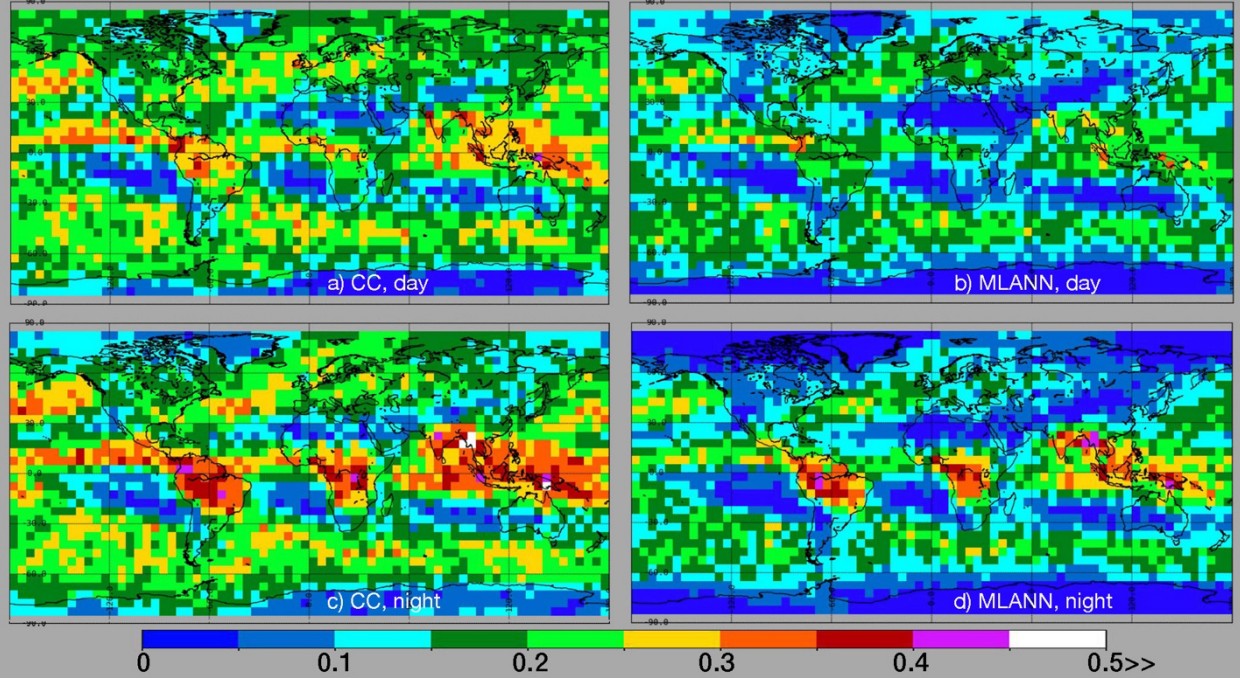

**Figure 4.** Fraction of matched 2009 CC and Aqua MODIS pixels classified as multilayer clouds. CALIPSO-CloudSat and Aqua MODIS ML classifications on left and right, respectively. Day and night pixels on top and bottom, respectively.







**Figure 5.** Zonal mean 2009 ML cloud fraction from matched CALIPSO-CloudSat and Aqua MODIS as in Fig. 4. Zonal differences, MLANN - CC, are also plotted. Global averages are indicated in the legend.



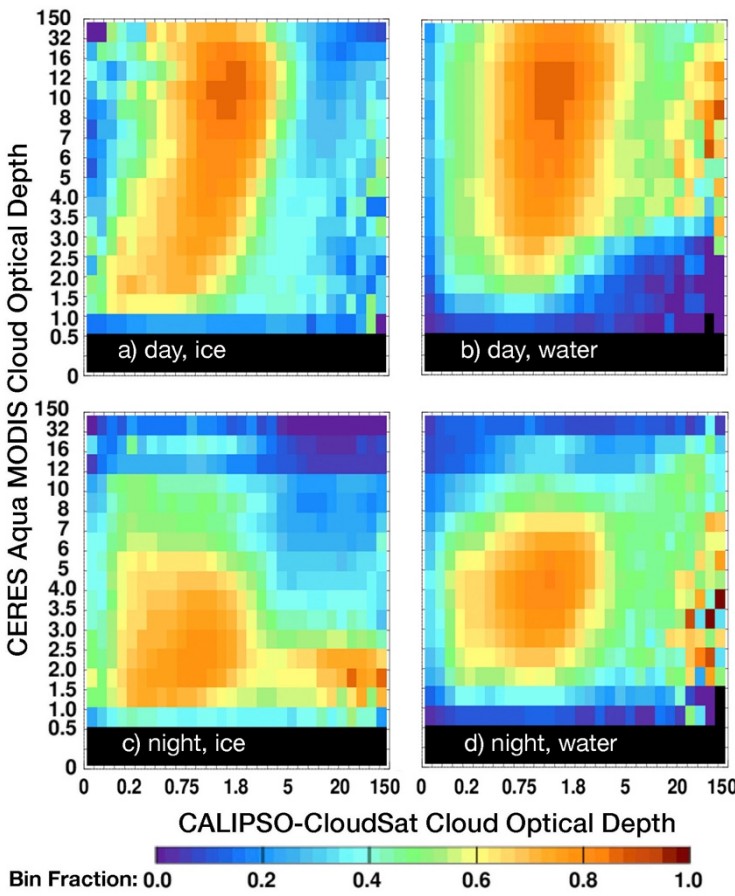

**Figure 6.** Recall or fraction of ML clouds detected within a given CC and CM cloud optical depth bin, 2009. Note, the irregular axis scales. The tick marks for the x-axis are 0, 0.0025, 0.05. 0.1, 0.15, 0.2, 0.3, 0.4, 0.5, 0.6, 0.75, 0.9, 1.1, 1.3, 1.5, 1.8, 2.1, 2.5, 3, 4, 5, 6, 8, 10, 15, 20, 30, 40, 60, 80, and 150.



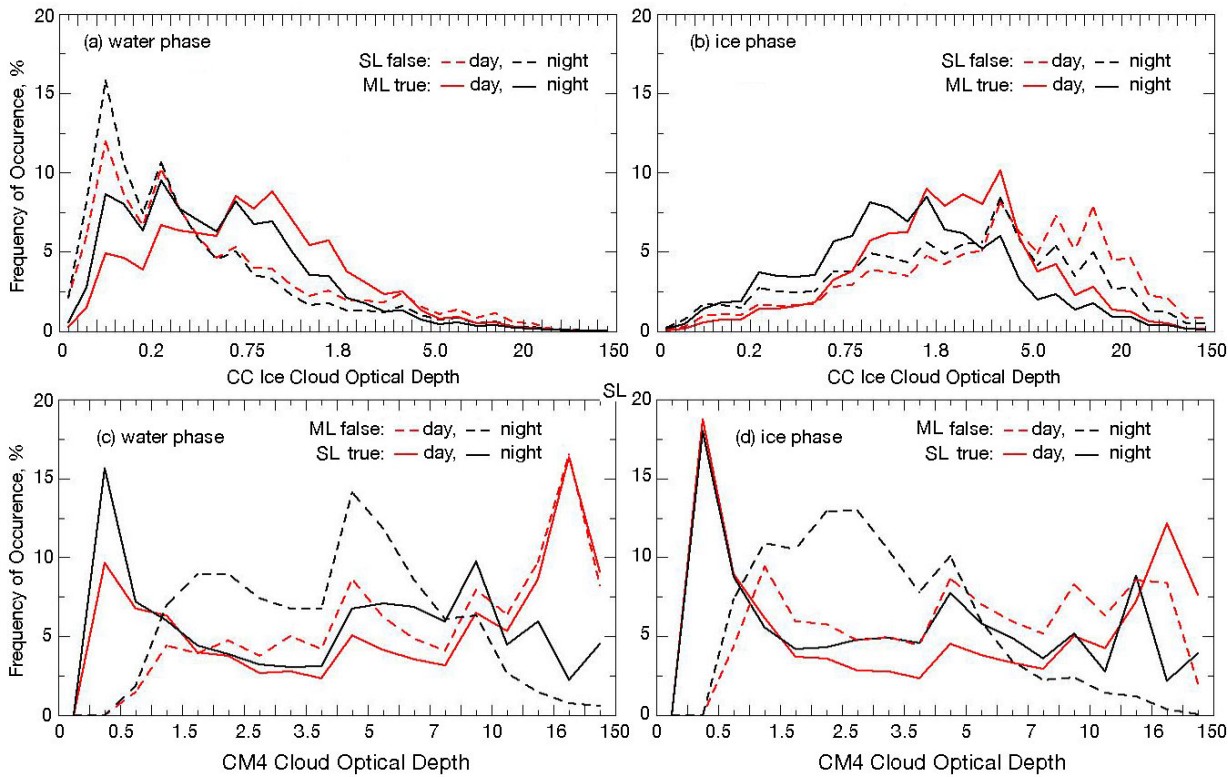

**Figure 7.** Probability distributions of 2009 false SL and true ML clouds from MLANN as functions of upper-layer cloud optical depth over SF surfaces for MODIS (a) water phase and (b) ice phase. Probability distributions of 2009 false ML and true SL clouds from MLANN as functions of total column cloud optical depth over SF surfaces for MODIS (c) water phase and (d) ice phase. The major tick marks for the x-axes on the top panels are 0, 0.0025, 0.05. 0.1, 0.15, 0.2, 0.3, 0.4, 0.5, 0.6, 0.75, 0.9, 1.1, 1.3, 1.5, 1.8, 2.1, 2.5, 3, 4, 5, 6, 8, 10, 15, 20, 30, 40, 60, 80, and 150. The major tick marks for the x-axes on the bottom panels are 0, 0.025, 0.5. 1.0, 1.5, 2.0, 2.5, 3.0, 3.5, 4, 5, 6, 7, 8, 10, 12, 16, 32, and 150.



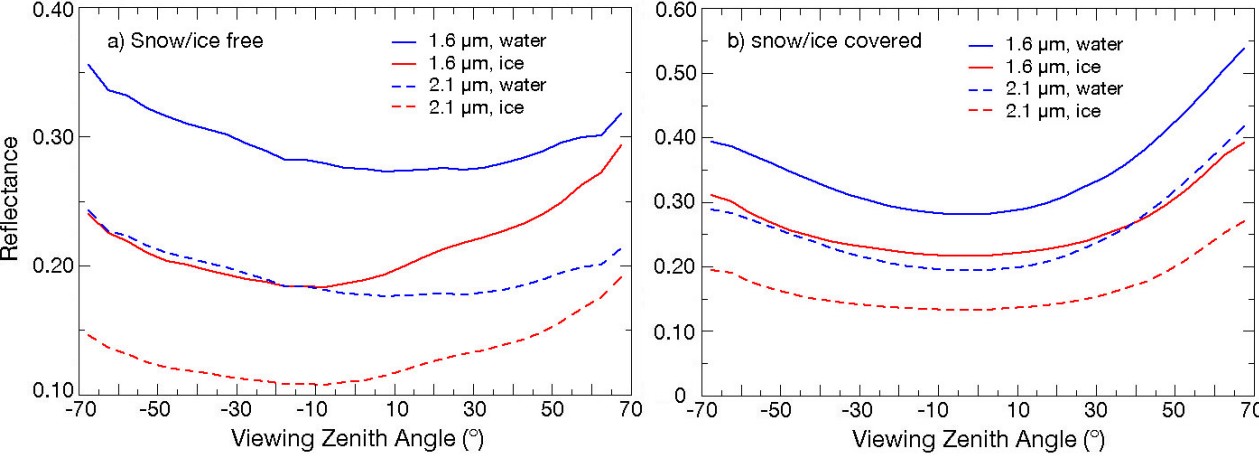

**Figure 8.** Mean reflectance from Aqua MODIS as a function of VZA for CERES water and ice-phase clouds, JAJO 2019.





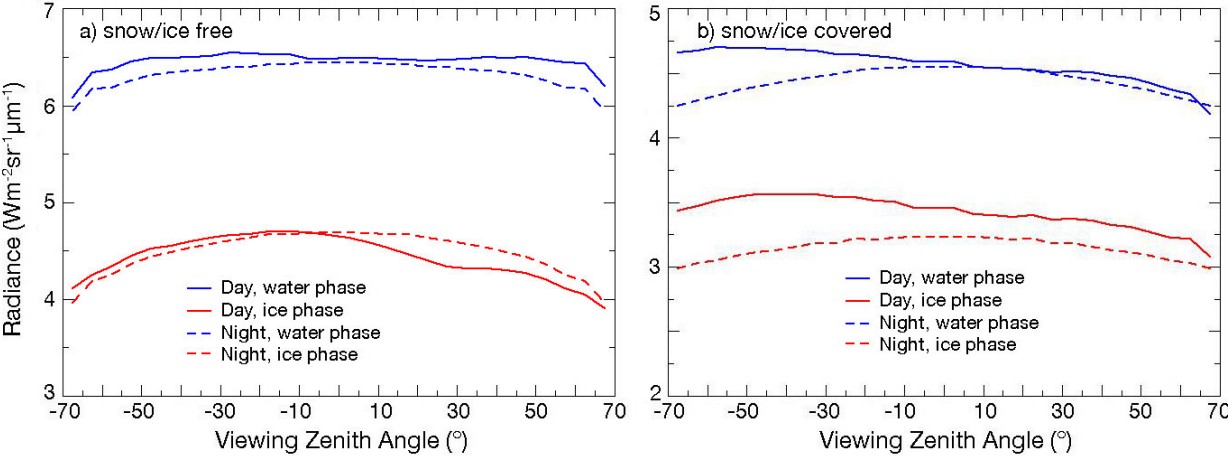

**Figure 9.** Mean 10.8-µm radiance from Aqua MODIS as a function of VZA for CERES water and ice-phase clouds, JAJO 2019.



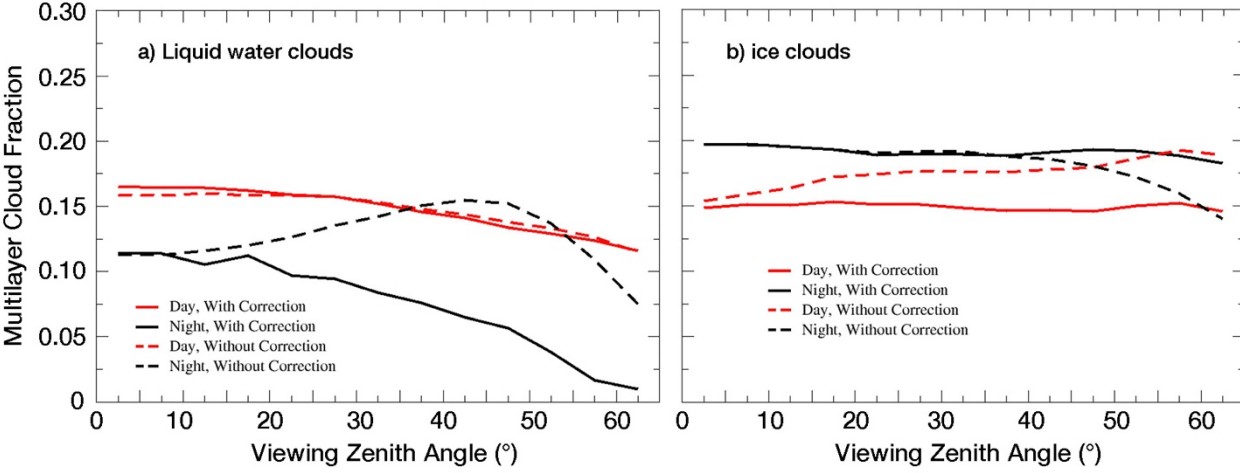

**Figure 10.** Mean MLANN multilayer cloud fraction from Aqua MODIS as a function viewing zenith angle, April 2009. MLANN

was run with the MODIS data as observed (without correction) and after applying a VZA correction (with correction).



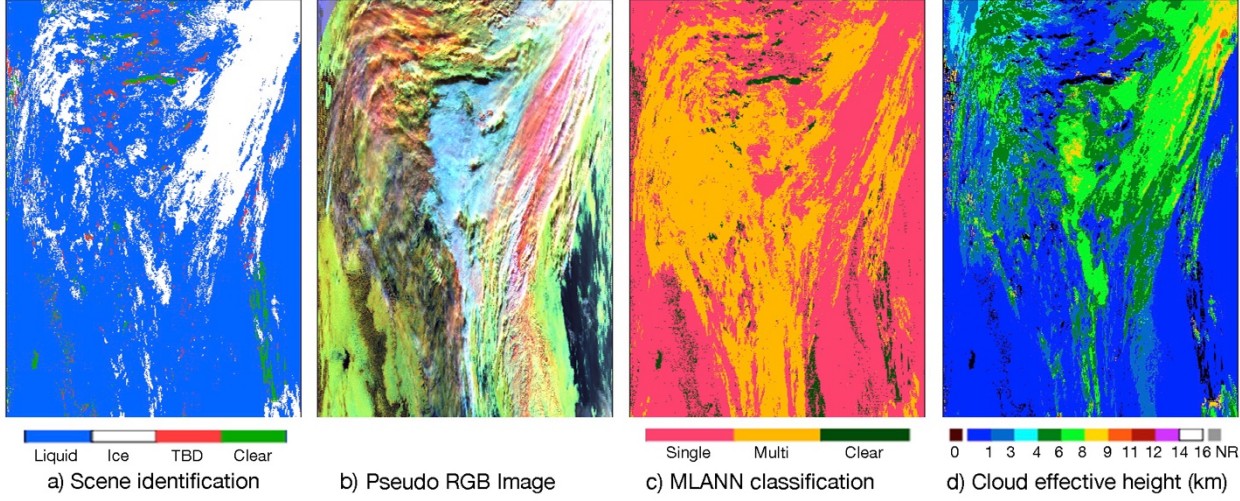

**Figure 11.** Cloud parameters derived from Aqua MODIS data between 62°S (top) and 52°S (bottom) around 165°E, at ~3:50 UTC, 16 April 2019. (a) CM4 pixel scene classification, (b) Pseudocolor RGB image, red: 0.64 µm reflectance, green: $BT_{37}$, green; blue: reverse $BT_{11}$. (c) MLANN classification, and (d) CM4 cloud effective height.



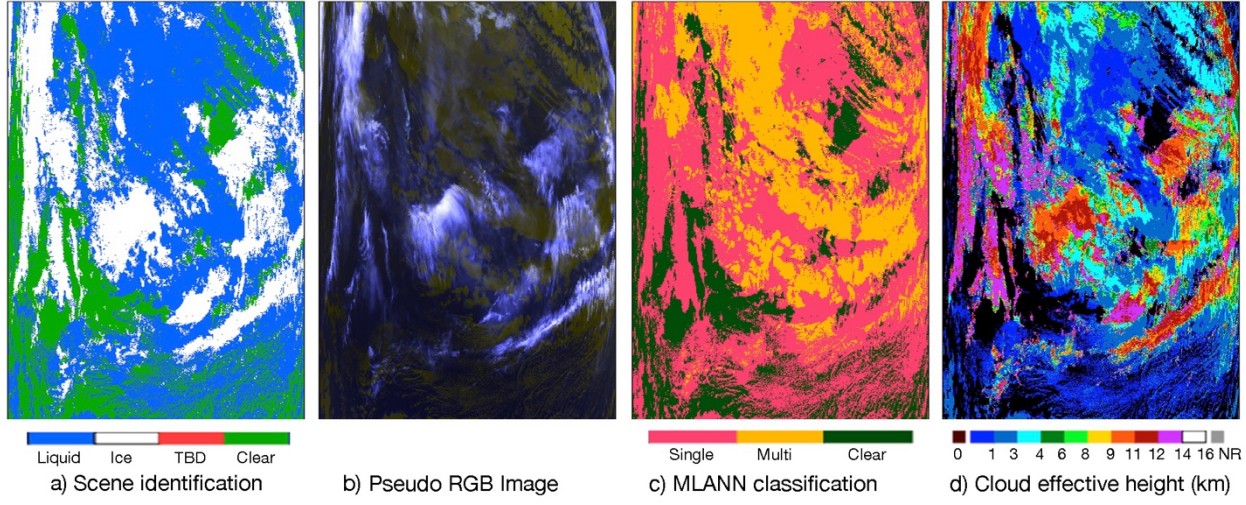

**Figure 12.** Cloud parameters derived from Aqua MODIS data between 42°N (top) and 24°N (bottom) around 50°W, at ~4:45 UTC, 16 April 2019. (a) CM4 pixel scene classification, (b) Pseudocolor RGB image, red: reverse $BT_{11}$, green: reverse $BT_{12}$, green; blue: $BTD_{3711}$. (c) MLANN classification, and (d) CM4 cloud effective height.




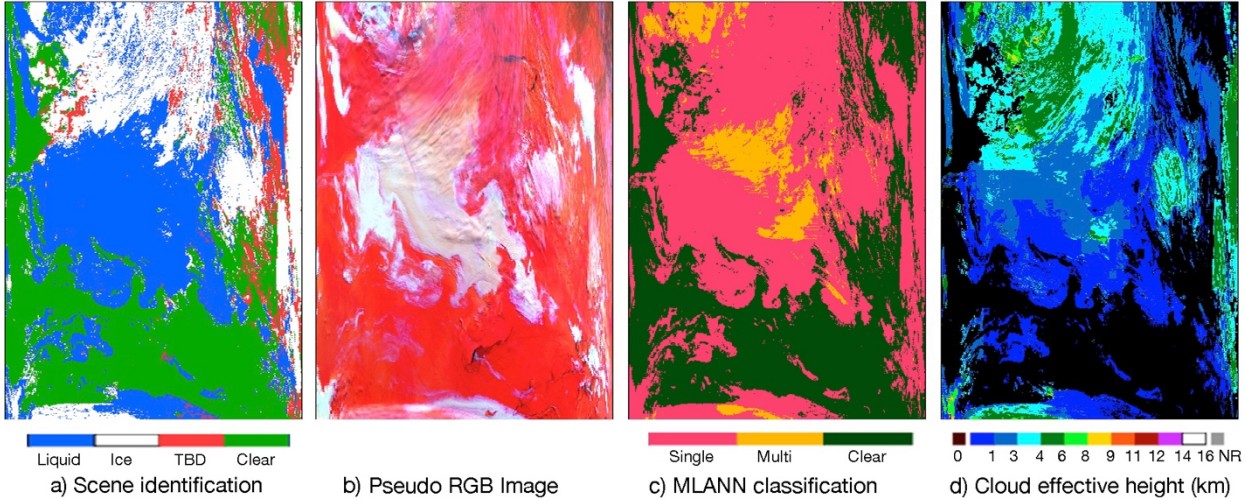

**Figure 13.** Cloud parameters derived from Aqua MODIS data between 77°N (top) and 83°N (bottom) around 155°E, at ~1:50 UTC, 16 April 2019. (a) CM4 pixel scene classification, (b) Pseudocolor RGB image, red: 0.64 μm reflectance, green: $BT_{37}$, green; blue: reverse $BT_{11}$. (c) MLANN classification, and (d) CM4 cloud effective height.





**Figure 14.** Multilayer fraction of total cloud cover for JAJO 2009 using all CC data from 2009 (top), and using Aqua MODIS MLANN retrievals (middle) at near-nadir (-18° < VZA , 3°), and (bottom) for all VZAs. Daytime on left, nighttime on right.





**Figure 15.** Monthly mean anomaly of multilayer fraction relative to total cloud cover for 2009 using all CC data and full-swath MODIS data.