# Peer review of "Identification of ice-over-water multilayer clouds using multispectral satellite data in an artificial neural network"

_EGUsphere, 2023_

## Author Comment (AC1)

Reviewer 1

The authors have submitted an interesting manuscript that presents the principle and building of a artifical neural network (ANN) based algorithm and its results. The objective is the identification of the multilayer character of ice-over-water multilayer clouds from MODIS/AQUA measurements. The ANN is trained with information from the active sensors of the A Train. While several approachs and attempts to retrieve such information about multilayer clouds from passive sensors exist, and they are listed in the manuscript and used for comparison, multilayer qualification is still a quite innovative retrieval target.

Authors give arguments about the importance to get this single layer (SL) and multilayer (ML) qualifications in order to improve the estimate of radiative budget, and to assimilate with improved confidence informations about clouds in different applications.

Data and methodology are clearly presented as well as the results. The analysis of the results are interesting : recall of ML cloud detected as a function of total and upper layer COD, probability distributions of true and false SL and ML clouds as a function of upper layer COD, attempts to correct and validate the use of the neural network as a function of the viewing angle in order to exploit it over the entire MODIS swath, presentation of case studies, demonstration with inferred climatologies of ML cloud occurrence of the performance and value of the approach, operationnal considerations.

Overall, the performance of the algorithm appears robust, consistent and, even if the comparison is not so straightforward. the performance's scores seem higher that those of other algorithms. The statistical bias of the algorithm appears to lie between -5 and -10 % (ML occurrence), bias that disappears when studying monthly anomalies following the fact that the bias is latitude independent and certainly, not shown, timely independent. Two very valuable aspects of this algorithm are that is performs during both day and night and also above snow-covered surfaces.

This work and its results fit well in the scope of EGUsphere, topic Atmospheric sciences and AMT.

As a reviewer, I will call here for minor revisions of this manuscript.

Thanks for reviewing this paper and providing constructive comments for improvement. We hope the concerns have been addressed satisfactorily.

Some omissions bring some lack of clarity.

First, the neural network algorithm appears here a little bit as a black box. It is understandable, as it is, in a way, a black box, but some descriptions could have been turned differently.

The usage, two times, of 'artificial intelligence', to describe its operation, participates in this shortcut. The ANN works here thanks to adequate choice of inputs and outputs and trainings. So it is not by itself intelligent.

It is true that the ANN is not by itself intelligent. However, the artificial neural network is recognized as a form of artificial intelligence, albeit a somewhat primitive one having first appeared in 1954. However, to alleviate worries that the MLANN might be intelligent unto itself, we have eliminated the terms from this paper.

Authors don't give two much technical details about how they design for their usage the NN, letting the reader find it in previous studies as Minnis et al (2016, 2019) and Sun-Mack et al (2017). And this technique seems to be in the continuity of the works of Sun-Mack et al (2017) and Minnis et al (2019). The current manuscript makes numerous times reference to these two previous communications. The first one is a conference paper, not necessarily a problem, but the communication is not very long and detailed. More problematic, the second one, Minnis et al (2019) is not given in the references 'page of the paper. So one should accept all what is said about this reference (and there are some lines as on lines 93 to 98, etc.) without having the possibility to read it … The reference should be given.

We have rewritten the Data and Methodology sections. Hopefully, this will clear up some of the problems. References have also been included.

Finally, the (too?) short description of the physical basis behind the usage of the ANN leads to some lack of clarity :

- concerning the contraint used for the training (top of page 8), the rationale is not given : why no temperature inversion between 273 and 253 K ? the rationale should certainly be added in order to better understand the applied filtering and the definiton of what are here selected as multilayer cloud situations, that would help in the appreciation of the difference between the presented results and the ones of the others references (MYD06 C6, POLDER, …)

The second condition has been rewritten as follows. We hope that clarifies the reasoning.

"(2) at least, one layer with extinction occurs at a height above the altitude corresponding to 253 K and no temperature inversion exists in the atmospheric layer between the altitudes corresponding to 273 K and 253 K. This constraint is used to eliminate the possibility of warm clouds occurring above the assumed ice threshold of 253 K."

- It is said on line 325 that 'The results represent a significant improvement over the previous formulation. Much of the increased accuracy is due to …' but the rationale of it could be clearer.

We have rewritten that paragraph as follows. We hope it is more understandable now.

"These results represent a significant improvement over the previous MLANN formulation (Minnis et al. 2019), which only attained accuracies of 80.4% and 77.1% during the day and night, respectively over SF surfaces. Much of the increased accuracy is due to use of shorter CALIPSO horizontal averaging distances here. By employing CALIPSO averages over distances up to 80 km, Minnis et al. (2019) attempted to detect ML cloud systems that included many cirrus clouds having optical depths smaller than 0.2. Such clouds are difficult to detect with passive remote sensing even when they are single-layered. According to Yost et al. (2021), systems having $\tau_{CC} < 0.2$ account for ~42% of all ML clouds for CALIPSO data using HA $\leq$ 80 km compared to only 18% for HA $\leq$ 5 km. A majority of those low-optical-depth ML clouds were not detected in Minnis et al. (2019), resulting in lower accuracies. Typically, cloud identification or multilayered cloud detection methods that use CALIPSO for validation or training employ data with HA $\leq$ 1 or 5 km (e.g., Desmons et al., 2017; Marchant et al., 2020; Tan et al., 2022; White et al., 2021). By using that smaller averaging distance in this study, the fraction of CC ML clouds is ~25% less than that used by Minnis et al. (2019), but a larger portion of them is detected. Other sources for the improvement arise from utilizing additional input parameters, including those based on the 13.3-µm channel and $\rho_{1.38}$, and $\rho_{1.61} - \rho_{2.13}$. Additionally, the assumption that all pixels having $\tau_{CM} < 0.5$ are automatically SL, regardless of the CALIPSO classification, probably removed some difficult but less important cases."

- On line 332 : 'improvements arise from using additional input parameters, including [IR and NIR] channels'. Also on line 333 : 'vertical profile of relative humidity'. Wasn't there a way to argue or illustrate more clearly, if not demonstrate, these added values ? (why and how much the vertical profile of relative humidity adds useful information ?).

Please see the 3rd paragraph in section 3.0 where we specify the contribution of the RH profiles.

Minor Questions :

- a basic question : why the choice of 50-70 neurons for the hidden layer is a good choice ?

See the addition to the first paragraph in the Methodology section.

"Only one hidden layer is used for this shallow neural network. It was found that a second layer yielded no significant increase in accuracy, but greatly increased processing time. The number of neurons in the hidden layer vary from 50 to 70 depending on the data category (e.g., snow-free daytime ice clouds). The exact number was determined by adding neurons until gains in accuracy ended."

- Concerning the choice of defining a cloudy pixel as ML when the MLANN output probability is higher than 0.5 :

What is the consequence of the threshold's choice ?

The accuracy decreases for values < or > 0.5 for ice < or > 0.55 for water. The difference in accuracy between using 0.5 or 0.55 as the threshold for water is 0.1% relative to the value at 0.50. Thus, we use 0.5 as the threshold for both. The paragraph in the new methodology section summarizes our reasoning (see next answer).

Have you thought about defining instead as output a probability between 0 and 1, as in Desmons et al (2017), with a definition of a binary threshold that would maximize the algorithm accuracy ?

We changed the statement to ,

"Output from the trained MLANN is a probability between 0 and 1 for each pixel. The latter value denotes certainty that the pixel includes ML clouds as defined here. For practical purposes it is necessary to select a threshold probability above which a pixel is designated as multilayered. A threshold value of 0.5 was chosen based on analysis of the accuracy of the results for probabilities between 0.3 and 0.60. The accuracies (risks) were found to be greatest (least) for thresholds between 0.50 and 0.55."

- a distinction is made between snow-free and snow-covered surfaces, which makes sense. Wouldn't it have been interesting to present performances with a distinction ocean/land ? (that would have shown a better performance during the night over land?

Training for land and ocean separately may gain additional accuracy, especially over land. However, one of the input values is surface type, land or ocean, which should lead to optimal performance over each surface. We can examine that in future studies.

- on line 270 : if CoS is called Single-layer Confidence, why not defining for consistency on line 268 PR as Multilayer confidence (CoM) ?

This is one of the terminology issues that has driven us and apparently Reviewer 2 to distraction. Desmons et al. (2019) used CoM, while Tan et al. (2022) used precision. We opted to use the Tan et al. (2022) nomenclature except for parameters they did not include.

- on line 270 : isn't the definition or equation wrong ? Isn't it instead False SL rate ? Or SS/(SS+SM). Values given in Tables seem correct ; but the definition is certainly wrong here.

Yes, the formula was not written correctly and has been revised.

- on line 511 : why is it 'not shown' ?

We were trying to keep the plots simple. We now show the full -70 to 70°. It is true they were not cancelling each other. The ice and water curves were mistakenly used to make the original statement. Thanks. The paragraph has been changed accordingly.

- From the results of Figure 5 : would there be an interest to plot the difference (night minus day) and show the capacity of MLANN to get right this difference ?

We have plotted it and found $R^2 = 0.88$ and included it as the new Fig. 5.

Minor comments or typo :

- line 88 : does the reference to Minnis et al (2023) exists ? It should be given.

Done

- line 108 : Venetsanopoulos

Not sure what is needed here. The name is spelled the same as in the paper.

- line 120 : equation with equal sign not to be cutted in two

Done

- line 315 : Table 3 instead of Table 2

Done

- sentence on line 368 and 369 : is it really two conditions on \tau_{CM} ?

Good catch. We corrected the second condition to reflect tauC, instead of tauCM.

- line 444 : Desmons instead of Desmond

Done

- line 446 : Marchant et al (2020) instead of (2017) ?

2020

- on line 573 : Fig. 14 instead of Fig. 16

Corrected

- line 613 : decreases from 87 %?

"from" inserted

- line 693 : Sourdeval instead of Sourdevall

Corrected

- on the legend of Fig. 3 : The acronyms SF and SC could be detailed

We have included the details in the caption.

- in Table 3 : on column 9 : some totals are wrong :

Thanks for the tips. We have made the corrections.

94.4 instead of 69.5 on line 1

5.7 instead of 30.5 on line 2

90.9 instead of 71.3 on line 5

9.1 instead of 28.7 on line 6

---

## Author Comment (AC2)

Reviewer 2

The submitted manuscript describes research towards an operational retrieval of vertical cloud structure from space-borne, imaging spectroradiometers. It continues development, by the same research team, of the MLANN algorithm: the present iteration 1) further subsets the algorithm (which already separately models day and night observations) to separately model observations from snow-or-ice covered and snow-or-ice free surfaces, and 2) uses updated labeled datasets. For the first time, the authors also consider the application of MLANN (which is trained on near-nadir viewing angles) to full swaths (i.e. including far-from-nadir viewing angles).

The manuscript is lengthy and often poorly organized, making a comprehensive list of revisions, sufficient to merit publication, too time-consuming to generate. I've listed several of my key concerns with the manuscript below, along with several examples of unnecessary additions to the manuscript's "cognitive load" which, if reduced, would allow this reviewer to complete a comprehensive list of recommendations.

We appreciate the reviewer's suffering through the slog and providing us with some very good recommendations. We hope that the changes are satisfactory and make the trek a bit smoother.

1. The discussion begins with an unsupported statement: "These results represent a significant improvement over the previous MLANN formulation." It's important that a revised manuscript include evidence supporting this conclusion. If this iteration of the MLANN has not shown improvements, then an entirely different manuscript crafted around a negative result is needed. While I'm sure that is not the case, where is the comparison? Section 5.2 and Table 5 have neglected that critical comparison.

The accuracies of the 2019 MLANN were originally reported in the penultimate paragraph of the Introduction. Apparently, the connection between Discussion statement and an implied comparison of the accuracies in Table 3 with those for 2019 in the Introduction was too subtle. To reduce cognitive distress, the discussion now begins with a more explicit supported version of that statement, as follows.

"These results represent a significant improvement over the previous MLANN formulation, which only attained accuracies of 80.4% and 77.1% during the day and night, respectively over SF surfaces."

2. Sections 2 and 3 are not laid out in a way that helps the reader. ANNs are data-driven, so presenting the model before the data demands the reader maintain abstractions "x" and "y" until the next section. I recommend reversing the presentation, so the reader knows the inputs and outputs. The "Input and Output Layers" subsection of 3 includes aspects of methodology (data splits, number of models trained) and feature definition (cloud layer classification) that don't have anything to do with the subsection heading.

Thanks for the comment. We have rearranged and rewritten the data and methodology sections.

3. **The accuracy metric featured foremost in the abstract and conclusion is sensitive to imbalance in the test data, which is nearly 80:20 in this manuscript. The null model for an 80:20 split also already has 68% accuracy (not the 50% naively assumed for a binary classification problem). You might consider using instead the Mathews correlation coefficient. You definitely do not want to compare, as in Table 5, the accuracy of models with different degrees of imbalance in the test data.**

We have added in the F1 score, which is also valuable for unbalanced binary data. It was also employed by Tan et al. (2022) and others (e.g., Haynes et al., 2022), and is easily computed from the data of Desmons et al. (2019). We believe it is a necessary exercise to perform the comparisons in Table 5 to put the MLANN results in context. The differences  among the methods and datasets are clearly outlined in the text. We prefer to leave it to the reader to gain a sense of how well the MLANN compares to other published methods.

4. Terminology and acronyms are abundant and sometimes more confusing than helpful. One example: does "multi-layer" in MLANN refer to neural network layers or cloud layers? MLNN is a widely used acronym for neural networks with multiple layers of nodes, so most readers will associate the ML in MLANN with nodes not clouds.

As to the choice of the acronym, we have to disagree. The acronym is clearly defined and contains "A", a distinct difference from MLNN. We expect that most readers of this paper are focused on the multilayer (ML) cloud component; multilayer clouds are in the title. We doubt there are that many readers of AMT familiar with MLNN as a reference to neural network layers. As a test of that contention, we looked at the 30 most recent AMT papers containing neural network in the title and none of them contained the acronym MLNN. We have removed the parameter, fraction correct or FC, to reduce the acronym load a bit. Its inclusion with its equivalent, ACC, was meant to make easy reference to the previous studies, but only added to the load.

A second example: the MS, SM, MM, SS acronyms for the confusion matrix are unnecessary departures from the true positive (TP), false positive (FP), etc. terminology widely used for binary classification problems. You could eliminate a whole Table by ditching MS, SM, MM, and SS in favor of conventional terminology.

Having slogged through much of the literature on using machine-learning methods for clouds ourselves, we encountered a variety of variable names employed for the confusion matrix. It is very confusing. We found that the MS, SM, MM, and SS set of Tan et al. (2022) was straightforward and explained at a glance what each number meant. We thought it would be suitable for multilayer detection. Apparently it is not suitable for everyone. Having gone back and forth on this, we opt to retain this terminology and Table 1 to ensure clarity. We have clearly defined what each variable is and find that it no more of burden than trying to remember if TP refers to ML or SL agreement..

A third example: what is the purpose of inventing "net gain of accuracy" when recall (of the multi-layer class) already quantifies MM with respect to MS and is both familiar and normalized?

Good question. The goal of this algorithm development is to detect as many multilayered clouds as possible while minimizing the number of false ML clouds as we stated in the text. Why? Certainly, not for the sake of doing it. It is simply the first step in characterizing the clouds within a column in terms of their macrophysical (vertical structure: height, thickness, layering), and microphysical and optical (phase, particle size, optical depth) properties. Having to concoct ML clouds and their properties out of whole cloth for a false ML pixel, when there is only one layer present, is certainly as much of an error as identifying a ML cloud as a SL cloud, the default case.

Because recall is a normalized parameter, it can yield the same value for different levels of multilayer cloud detection. Similarly, its counterpart, precision, is also normalized. As such, it is the same whether MM = 3 and MS =1 or MM = 30 and SM = 10. It does not tell us, to use an American football analogy, whether we moved the ball 2 yards or 20 yards toward the goal. We could not find an appropriate straightforward metric from among the standard parameters used in confusion matrix analysis, so we defined one, net gain in accuracy, NGA, for the purposes of this analysis. It represents the net effect of the effort. Additionally, it serves as a means of comparison with some other methods that may use different unbalanced test data. It is the only new parameter that we have introduced.

A fourth example: CoS is a new term and acronym for precision of the single-layer class. (At least, it is if I'm correct that the "1 - " in equation 6 is a mistake.) Every new term or acronym, especially ones close to but deviating from familiar ones, increases the cognitive load the reader must maintain while interpreting the study.

You are correct, the "1 -" was a typo. CoS is not our term, but was used by Desmond et al. (2017). It was employed to facilitate comparisons with other algorithms as stated in the text and seen in Table 5. It is much more descriptive of the quantity than one of the standard confusion matrix terms.

5. The presentation of the algorithm spends too long reviewing general theory and barely touches useful specifics. I appreciate brevity allowed by references to prior papers documenting the MLANN, although I did not take time to read them too. Nevertheless, specifics that either differ, are essential for reproducibility, or simply quick to convey ought to be included (e.g. exact network shape for each model, the loss function). The description also has to be self-inconsistent, which it is not in describing 1) how "testing" and "validation" data are (both?) used to terminate training, and 2) describing y as a probability in the text but as any real number in Figure 1. One thing the cited papers may explain, but I would like to (also) see here, is the reasoning or optimization that 1) limited the ANN to a single hidden layer, and 2) allowed collinear variables as inputs (in the brightness temperatures along with brightness temperature differences).

We have rewritten the methodology section to eliminate generalities and  provide specifics. Please see the revision.

 The reasoning for using a single hidden layer is included in the revision:

"Only one hidden layer is used for this shallow neural network. It was found that a second layer yielded no significant increase in accuracy, but greatly increased processing time."

The reasoning for including the collinear variables is simple. Their inclusion increases the detection accuracy. Others using machine learning methods have come to the same conclusions (e.g., Tan et al., 2022; Hakansson et al. 2018). As a measure of that improvement, we have added the following paragraph to the methodology section.

"The input variables in Table 2 were selected by adding in parameters suspected of enhancing ML detectability and computing accuracies for each run. If no gain in accuracy occurred, the parameter was not used. Each predictive parameter's influence on the final MLANN formulation was assessed by computing the relative decrease in recall when a given parameter was removed from the training. Recall is the fraction of the CC ML pixels that were detected by the neural network.The decrease for each parameter was divided by the sum of all of the values to produce a relative ranking of importance. The ranks ranged from 0.038 for $BT_{11}$ to 0.082 for the relative humidity profiles, which were treated as a single input for these purposes. The second highest ranked parameter is latitude, followed in the daytime by SZA and $\rho_{1.38}$. In general, the brightness temperatures were ranked lower than the BTDs, similar to the rankings reported by Tan et al. (2023) for their random forest method."

6. The optimization appears to take not precautions against local minima or overfitting. Widely used software for neural network optimization use some form of stochastic gradient descent, whereas this study uses Levenberg-Marquette. How are local minima avoided? The large numbers of parameters in neural network optimization can lead to overfitting, but the authors do not indicate any steps taken to avoid overfitting or indicate that none was observed.

Thanks for pointing out our omissions. In doing so, you enabled us to catch an error. We used the Levenberg-Marquardt function in the previous MLANN version and forgot to replace it in the current text with the scaled conjugate function. We now explain our approach to avoiding local minima and and overfitting. Because we did not use a deep-learning network, we did not employ the SGD. Instead, we took a "manual" approach, as described in the new paragraph:

"To avoid local minima in the neural network, the training runs were repeated many times using different samplings of the dataset (e.g., every 3rd pixel or every 5th pixel); different random initial weights; and various percentages for training, testing, and validation. Local minima were identified when the training convergence time was abnormally short or long. Overfitting was avoided by using a very large dataset (typically more than a million datapoints), which forces the net to generalize. It was also avoided by using a minimal number of neurons. Additionally, unreasonable data, such as fill values, were filtered out to minimize the noise. A set of range limits was used to eliminate any obviously errant data. Leaving such data in the input set prevents the training from generalizing. Unnecessary input parameters were also removed by trial and error to streamline the training. Finally, similar performances of the MLANN with the 2008 training and 2009 independent validation datasets ensured that the trained network was producing global minima without overfitting."

7. The function "g" in Figure 1 is a shifted and scaled sigmoid activation function, but neither the shifting nor scaling will have any effect given the free weights and biases. This unnecessary "tweak" is another addition to the cognitive load that makes reading this manuscript a real slog.

Figure 1 appears to have caused nothing but confusion. To ease the cognitive overload, it and all references to it have been removed from the revision.

8. The research ought to be reproducible. Normally I would say "more easily reproducible", but this work has not reached the bar of reproducible at all. The software and data ought to be provided, preferably coupled with a clear pipeline for training and evaluating the model. Software the authors didn't write must be cited (what software evaluated and optimized the neural network?), and any software the authors wrote ought to be included as a supplement. If a pipeline includes downloading and processing of raw data from some CERES Ordering Tool API, then the link-to-the-data provided in the manuscript is sufficient; if it does not, the processed data ought to be published.

The Mathworks software that we used is now referenced and linked. With that Matlab package and the C3M data linked at the end, it should be possible to reproduce any of the training statistics.

9. Minnis et al. 2019 is not in the references.

It has been added. It could also have been made available upon request.

10. The subscript on f in Figure 1 is not needed. In fact, it is a misdirection because the subscript indicate the existence of hidden layers which do not exist. The nodes currently labelled f_1 should be labelled u_1, u_2, u_3, etc. What is the difference in Figure 1 between x_1, x_2, x_3, etc. and the first layer of nodes with no label?

Figure 1 is no longer included.

11. Line 207-209 is an incomplete sentence.
Corrected

12. The studies included in table 5 do not use the same test data, so comparing the metrics is okay but not entirely quantitative. Indicating the "best" result with a bold-face font pushes the comparison too far.

Bolding removed.

13. The similarity of the CC and MLANN cloud fractions when stratified over space (Fig 14) or time (Fig 15) could be quantified with a correlation coefficient.

We have added the following to the paragraph describing day Fig. 14 results:

"Linear regression between the daytime CC and the MLANN regional means yields $R^2$ values of 0.80 and 0.66 for the near-nadir and full-swath results, respectively. The smaller value for the full-swath data is not surprising given its greater sampling. For the matched near-nadir and full-swath means, $R^2 = 0.81$."

We added the following to the paragraph describing the nighttime Fig. 14 results:

"The correlation coefficients are 0.71 and 0.64 for the nocturnal CC regional means matched with their respective MLANN near-nadir and full-swath counterparts, while $R^2$ is 0.89 for the matched near-nadir and full-swath averages."

We added the following to the paragraph describing the Fig. 15 results:

"The values of $R^2$ between the CC and MLANN monthly means are 0.92 and 0.90 for day and night, respectively."

---

## Author Response (AR2)

12 April 2024

To: Andrew Sayer, AMT Editor

Dear Sir,

We have reviewed the helpful comments provided by Reviewer #2 and have made a number of changes where appropriate. Those changes are noted in our response to the reviewer and can be seen in the revised manuscript, which retains the tracked changes. We have also made several zip files of the output of our analyses so that anyone interested in reproducing the results will have a reference. The link to the zip files is given along with raw data and Mathworks in the relevant section after the conclusions and summary.

We contend that we have gone far and above the efforts of previous studies of multilayer clouds to assess the results and provide a realistic picture of what the algorithm actually does. To our knowledge, no previous paper has explicitly accounted for the what the false detections actually mean in terms of the number of multilayered clouds that are detected by a given algorithm. In other words, instead of ignoring the impact of the false detections, we have hung out our dirty laundry with the NGA parameter and the revised plots in Figs. 3 and 4. We hope that these examples will set a new standard for reporting of multilayer cloud algorithm results in future publications.

We hope that you find the responses and changes satisfactory for publication.  Thanks for your consideration of our revised manuscript.

Sincerely,

Sunny Sun-Mack

Response to Review of first revision. Blue indicates our written response to each topic. Red indicates the text we have placed in the new revision.

> We appreciate the reviewer's suffering through the slog and providing us with some very good
> recommendations. We hope that the changes are satisfactory and make the trek a bit smoother.

The changes introduced are all for the better, and the trek is, just as advertised, a bit smoother.

The revised manuscript provides a rich description of successful modeling efforts that will allow departures from the single-layer, plane-parallel cloud morphology assumption to improve cloud property retrievals critical to CERES and other applications. The authors have not made a serious effort to quantify the nature of the improvement against the previous version of MLANN, but this is only one part of their report. The general performance assessment, comparisons against related works, and investigation of full-swath retrievals reflect mostly sound analyses and support the stated conclusions.

Evidence for improvement against the previous version of the MLANN is only quantitatively described by total accuracy. This reviewer has raised concerns about the utility of the accuracy metric for unbalanced classes. The authors the say that "much" of the change in accuracy is due to the new dataset, which gained increased imbalance due to the shorter CALIPSO horizontal averaging. A null model also has better accuracy if you increase the imablance in the dataset. Rather than a qualitative attribution of model improvement to the class being easier to guess blindly, this reviewer would like to know if the model is better because of improved architecture or the increased subsetting (from 2 trained models to 8 trained models).

Assuming the reviewer did not read Minnis et al. (2019), as was indicated in the first review, the reviewer must have missed the point of that formulation, which was applied only to SF surfaces and had separate training for ice and water clouds (4 models). The only change here was the addition of snow-covered surfaces, which added 4 more models to the mix. We have to admit, however, that he/she was led astray by our statement in the last paragraph of the Introduction, where we errantly noted,

"*In addition to the **separate day/night training used in previous versions**, the MLANN herein is trained separately for both snow-free and snow/ice-covered surfaces using an entire year of data.*"

That was a leftover from a cut and paste that we apparently overlooked. The separation of ice and water was a step up from the formulation of Sun-Mack et al. (2017), which only had 2 models for SF surfaces, day and night. Thus, Minnis et al. (2019) increased the MLANN accuracy with the use of the separate cloud phases. To make the progress of the MCANN, nee MLANN, clearer, we changed the last two paragraphs of the Intro to read:

"To improve the CERES ML detection, Sun-Mack et al. (2017) developed a multi-layer cloud detection ANN (MLANN) to distinguish between SL and ML clouds using MODIS radiance data matched to CALIPSO and CloudSat vertical profiles of clouds over snow-free surfaces. The MLANN was trained separately for day and night data. Minnis et al. (2019) enhanced the MLANN by including more input parameters and additional output variables such as upper layer CTH, COD, and cloud-base height (CBH). They also used only high-confidence CloudSat and CALIPSO data for training and further trained the MLANN separately for clouds identified as either ice or water phase by the CERES CM4 algorithms. Using one month of data, they found that, for nonpolar clouds, the MLANN correctly identified ML and SL clouds together 80.4% and 77.1% of the time during day and night, respectively, using CALIPSO data averaged over an 80-km distance. Those values are 5% greater in absolute terms than their earlier counterparts. While the accuracies are quite encouraging, the approach needs further refinement and complete seasonal and global coverage.

This paper reports on continued development of the MLANN to detect ML clouds. Revisions to the previous training are made using newer versions of CALIPSO and CloudSat products with constrained horizontal resolution. To be more representative of its use and avoid confusion with other machine-learning terms, the acronym for this revision is changed from MLANN to the Multilayer Cloud-detection Artificial Neural Network, MCANN. To expand coverage to the entire globe, the MCANN is trained separately for CERES ice and water cloud pixels separately for snow-free and snow/ice-covered surfaces using an entire year of data. Input to the MCANN is also enhanced with some new variables. Finally, because the MCANN is trained with near-nadir data, its utility for full swath MODIS data is examined."

The authors suggest that the F1 score handles imbalance, which could be true depending on details of scoring, but do not apply even that in their comparison. I encourage the authors to better quantify the improvement in the MLANN, and consider the subtleties of the F1 score (in particular, consider the differences between what scikit implements as "binary", "macro", and "weighted" averaging [1]).

We have now included the F1 score as well as other parameters to further bolster our claim that the new version is more accurate than the previous one. The beginning of the Discussion Section now reads:

"These results represent a significant improvement over the MLANN of Minnis et al. (2019), which only attained accuracies of 80.4% and 77.1% during the day and night, respectively over SF surfaces **using a single month of data. Over SF surfaces, PR, RC, and F1 from the MCANN are 74% (72%), 57% (52%), and 64% (60%) for daytime (nighttime), respectively. In relative terms, all of those values exceed their MLANN counterparts by 1% to 20%. The MLANN NGA values are slightly higher.** Much of the increased accuracy **of the MCANN relative to the MLANN** is due…"

The slightly higher NGA values arise simply because there is a much greater population of ML clouds than available in the current study. We believe the numbers we have listed in the latest revision are sufficient. We have also provided the relevant confusion matrices in Minnis et al. (2019) and in this paper for anyone to make whatever calculations they wish in order to further assess the relative accuracies of these two methods.

The MLANN validation results (Table 4, Figures 3-4) emphasize two metrics, the accuracy and the multi-layer cloud (MLC) fraction. You tire of me complaining about accuracy, so I will only raise issue with the MLC fraction. The MLC fraction makes a lot of sense in Figure 14, where the CC labels are not available for a pixel by pixel comparison. When the correct classes are available, you actually know how much the MLC fraction is pumped up by false positives. Consider your statement (line 308) that MLANN underestimates the MLC fraction over snow-free surfaces by 4.8% during the day; actually, a whopping 4.1% of the classifications going into the fact were false positives so the correct MLC fraction was worse underestimate. The MLC fraction based only on correct classifications could be determined and presented in Figures 3-4, or a different aggregation I've not thought of would be better.

We are glad that the reviewer is seeing the point of NGA. What is net gain? All previous published maps and zonal means use the total number of ML detections as we do in Figures 3b, 3d, and 4. Plotting what is asked is unprecedented, but certainly logical, given that we have made a fuss about NGA. Therefore, we agree and have added two additional maps to Figure 3 and new points to Figure 3 that provide only the correct fraction of ML detections. We have accompanied those changes with relevant discussion.  We added the following to the penultimate paragraph in the Results section.

"Figures 3c and 3d show the ML fractions determined from all of the positive ML detections from the MCANN, whether they are correct or not. Figures 3e and 3f show the corresponding distributions of ML fractions based only on pixels that are actually correct according to the matched CC data. As expected from Table 4, the magnitudes for both

day and night are significantly reduced compared to those for all of the MCANN results. The relative distributions, of the correct values are similar to their all-MCANN counterparts."

We added the following to the end of the last paragraph of the results section.

"When only the correct MCANN values are considered (open squares), the zonal means drop further below the CC averages. The correct-MCANN differences relative to the CC values, shown at the bottom of Fig. 4, vary zonally much like their all-MCANN counterparts for both day (orange dotted line) and night (blue dotted line), but are 0.02 to 0.07 lower. The sources of these differences are discussed further below."

Table 4 is said to be based on the "training" data: I assume, but it's not explicitly clear, that this is the 20% test split.

Table 4 is based on the 2009 independent dataset.  Perhaps, Table 3 was meant here. If that is the case, then we have clarified it in the text and table caption.
For the former, the starting line of the relevant paragraph now reads, "*... The results in Table 3, based on the entire 2008 training data, include the confusion matrices...*".
For the latter, the Table 3 caption reads: "*... layer identification from the entire 2008 CloudSat-CALIPSO training set...*".

The authors acknowledge that comparison against related works is contextual, and not any kind of statistical ranking, because of the differences in the datasets. I agree, and also agree it is necessary to do. The final summary of this comparison (line 676 - 678) does not reflect the fuzziness of the comparison and should be more circumspect.

To overly emphasize this concern, we have inserted the following paragraph between the first and second paragraphs of the Conclusions section:

"Direct comparisons of the MCANN to other multilayer cloud detection methods are not possible due to differences in ML cloud definitions, input satellite data, reported accuracy parameters, sampling, and cloud optical depth constraints. Nevertheless, the MCANN results were evaluated here against published results based on other techniques. Attempts were made to minimize the characteristic differences among the various results as much as possible. All conclusions drawn from those comparisons are limited by the unknown effects of the remaining differences among the methods."

I again encourage the authors to re-consider their choices of acronyms. Calling the

model the multi-layer artificial neural network (MLANN) is vague, because the name doesn't include clouds, and may mislead readers already familiar with multi-layer artificial neural networks.

In the above reviewer comments, there is much concern that we did not elaborate on the change in the accuracy from our previous version of the MLANN. In so doing, it recognizes our previous use of the acronym, MLANN. That would suggest that we maintain the MLANN for continuity. However, the argument used in this second review for a change in acronym is much better than that from the first review. "Cloud" is nowhere to be found. Thus, we agree to change the acronym to MCANN, Multilayer Cloud-detection ANN. The statement noting that change is included in the answer to one of the questions above. MLANN is retained when referring to the previous versions, because they are already published.

Funny thing about the concern previously raised with respect to the SS, MS, MM, and SM acronyms: I mistakenly referred to "recall" when I meant "precision" in my complaint because I mixed up the unfamiliar MS and SM, and the authors also mixed up MS and SM in their reply regarding the NGA. I doubt either of us would have made mistakes with the usual TN, FP, TP, and FN.

The use of SM in the reply was a typo, not a confusion of the MS and SM. The revised paper stated it correctly. Given that we have defined our variables in Table 2 and our previous response, it should be quite simple for anyone to understand what we have done.

[1]: https://scikit-learn.org/stable/modules/generated/sklearn.metrics.f1_score.html